# Record-breaking dust loading during two mega dust storm events over northern China in March 2021: aerosol optical/radiative properties and meteorological drivers

Ke Gui[1], Wenrui Yao[1, 2], Huizheng Che[1, *], Linchang An[3], Yu Zheng[1], Lei Li[1], Hujia Zhao[4], Lei Zhang[1], Junting Zhong[1], Yaqiang Wang[1], Xiaoye Zhang[1]

[1] State Key Laboratory of Severe Weather & Key Laboratory of Atmospheric Chemistry of CMA, Chinese Academy of Meteorological Sciences, Beijing, 100081, China
[2] Department of Atmospheric and Oceanic Sciences & Institute of Atmospheric Sciences, Fudan University, Shanghai, 200438, China
[3] National Meteorological Center, CMA, Beijing 100081, China
[4] Institute of Atmospheric Environment, China Meteorological Administration, Shenyang, 110166, China

*Correspondence to*: Huizheng Che (chehz@cma.gov.cn)

**Abstract.** Although a remarkable reduction in the frequency of sand and dust storms (SDSs) in the past several decades has been reported over northern China (NC), two unexpected mega SDSs occurred on March 15–20, 2021 and March 27–29, 2021 (abbreviated as the "3.15" and "3.27" SDS events), which has reawakened widespread concern. This study characterizes the optical, microphysical, and radiative properties of aerosols and their meteorological drivers during these two SDS events using the sun photometer observations in Beijing and a comprehensive set of multiple satellite (including MODIS, VIIRS, CALIOP, and Himawari-8) and ground-based observations (including the CMA visibility network and AD-Net) combined with atmospheric reanalysis data. Moreover, a long-term (2000–2021) dust optical depth (DOD) dataset retrieved from MODIS measurements was also utilized to evaluate the historical ranking of the dust loading in NC during dust events. During the 3.15/3.27 event, the invasion of dust plumes greatly degraded the visibility over large areas of NC, with extreme low visibility of 50m and 500m recorded at most sites on March 15 and 28, respectively. Despite the shorter duration of the 3.27 event relative to the 3.15 event, sun photometer and satellite observations in Beijing recorded a larger peak AOD (~2.5) in the former than in the latter (~2.0), which was mainly attributed to the short-term intrusion of coarse-mode dust particles with larger effective radii (~1.9 μm) and volume concentrations (~2.0 $\mu m^3$ $\mu m^{-2}$) during the 3.27 event. The shortwave direct aerosol radiative forcing induced by dust was estimated to be −92.1 and −111.4 W $m^{-2}$ at the top of the atmosphere, −184.7 and −296.2 W $m^{-2}$ at the surface, and +92.6 and +184.8 W $m^{-2}$ in the atmosphere in Beijing during the 315 and 3.27 event, respectively. CALIOP observations show that during the 3.15 event the dust plume was lifted to an altitude of 4–8 km, and its range of impact extended from the dust source to the eastern coast of China. In contrast, the lifting height of the dust plume during the 3.27 event was lower than that during 3.15 event, which was also confirmed by ground-based Lidar observations. The MODIS-retrieved DOD data registered these two massive SDS events as the most intense episode in

the same period in history over the past two decades. These two extreme SDS events were associated with both atmospheric circulation extremes and local meteorological anomalies that favored enhanced dust emissions in the Gobi Desert (GD) across southern Mongolia and NC. Meteorological analysis revealed that both SDS events were triggered by an exceptionally strong Mongolian cyclone generated at nearly the same location (along the central and eastern plateau of Inner Mongolia) in conjunction with a surface-level cold high-pressure system at the rear, albeit with differences in magnitude and spatial extent of impact. In the GD, the early melting of spring snow caused by near-surface temperature anomalies over dust source regions, together with negative soil moisture anomalies induced by decreased precipitation, formed drier and barer soil surfaces, which allowed for increased emissions of dust into the atmosphere by strongly enhanced surface winds generated by the Mongolian cyclone.

## 1 Introduction

Sand and dust storms (SDSs) are a highly hazardous and disastrous weather type formed when strong winds draw large amounts of mineral dust aerosols from dry, bare soil surfaces into the atmosphere, and are a serious environmental problem that many countries adjacent to and downwind of dust source areas are or have been facing (Zhang et al., 2003a; Wu et al., 2021; Yu et al., 2021). East Asia is the world's second largest dust source, contributing about 40% (~8–13 Tg) of the global dust loading, with the largest contribution from the Taklimakan and Gobi deserts (TD and GD) located in northern China (NC) (Kok et al., 2021). East Asian dust can be transported to large parts of China (An et al., 2018), Japan, Korea, the Pacific Ocean (Zhang et al., 2003b; Tan et al., 2017), and as far as the west coast of the United States (Gong et al., 2006; Duncan Fairlie et al., 2007; Zhao et al., 2008), where it can affect the regional air quality, human health and many different socioeconomic activities, and modify Earth's energy balances directly by interactions with radiation and indirectly by interactions with clouds and ecosystems (e.g., Rosenfeld et al., 2001; Kok et al., 2017). Most SDSs in East Asia tend to occur in spring (March, April and May), and can mainly be attributed to two aspects: (1) the low cover of vegetation in spring, accompanied by low precipitation, leads to a dry and loose soil surface layer, which directly provides a favorable material source for the occurrence of SDSs; and (2) the frequent cyclones (mostly Mongolian cyclones) with strong northwesterly winds in spring combined with the unstable atmospheric stratification in the afternoon provide favorable dynamical conditions for the occurrence of SDSs (Chen et al., 2017a; Rodríguez et al., 2012).

Given the considerable environmental and climatic effects of dust aerosols, numerous studies have investigated the sources, spatial and temporal distribution, and long-term variability of spring dust aerosol loading in East Asia, and their drivers. Broadly speaking, these studies have revealed, from different perspectives, that dust emissions and loading as well as the frequency of SDSs in spring in East Asia have undergone a remarkable decline in the past several decades (e.g., Gong et al., 2004; Wang et al., 2018; Liu et al., 2020; Yao et al., 2021; Gui et al., 2021). The negative trends of spring dust aerosols in East Asia are mainly attributed to the decline in surface wind speeds, which may be related to the weakened temperature gradient at mid-latitudes caused by the enhancement of the Arctic amplification effect and weakening of the polar vortex (An

et al., 2018; Liu et al., 2020). In addition to the contribution from wind speed, the reduction in dust aerosols is also closely associated with the increases in vegetation cover (VC), precipitation (PPT), and volumetric soil water (VSW) in the dust source areas. Increased VC is not only driven by natural conditions such as temperature and PPT in mid- and high-latitude regions, but is often attributed to large-scale land-use management activities (e.g., afforestation in NC) (Chen et al., 2019).

In the context of a remarkable reduction in the frequency of SDSs in East Asia, and especially the absence of an SDS for
more than 10 years in NC, two unexpected extreme SDSs occurred on March 15–20, 2021 and March 27–29, 2021 (abbreviated as the "3.15" and "3.27" SDS events), both of which greatly degraded the air quality in most of China. Fig. 1 presents a snapshot panorama of the dust plume invading NC captured by the multi-spectral Advanced Himawari Imager (AHI) aboard the Himawari-8 geostationary satellite at 13:00 CST (China standard time) on March 15 and March 28, 2021, respectively. The satellite image on March 15 displays a dense dust plume that ravaged a large part of China, with an area of
more than 3.8 million $km^2$, accounting for about 40% of China's land area. Compared with March 15, satellite images on March 27 show a weaker dust plume intensity, along with a reduced eastward influence and scope. These two events received extensive media exposure as a result of their severity and enormous impacts on large areas. For example, both the World Meteorological Organization (https://public.wmo.int/en/media/news/severe-sand-and-dust-storm-hits-asia) and CNN (https://edition.cnn.com/2021/03/15/asia/beijing-sandstorm-decade-intl-hnk) described the 3.15 SDS event as the biggest
SDS in almost a decade. To date, several studies (e.g., Liang et al., 2021; Filonchyk, 2022; Filonchyk and Peterson, 2022) have been conducted to characterize the severe SDS event in March 2021. Liang et al. (2021) revealed the changes in dust composition and transport processes during the 3.15 SDS event in NC using geochemical analyses and remote sensing combined with backward trajectories analysis. Filonchyk (2022) and Filonchyk and Peterson (2022) preliminarily analyzed the synoptic conditions during the development of the 3.15 SDS event and assessed its impact on urban air quality using
particulate matter ($PM_{10}$) concentration observations. Also, the predominance of dust particles during the storm and the uplifted height of the dust plume were confirmed by using the Moderate Resolution Imaging Spectroradiometer (MODIS) observations and the Vertical Feature Mask (VFM) data from the Cloud–Aerosol Lidar and Infrared Pathfinder Satellite Observation (CALIPSO). These studies have focused on describing the transport processes of dust and its impact on air quality without elucidating in detail how atmospheric circulation patterns control the emission and transport of dust, and how
anomalous are these atmospheric circulation patterns and local meteorological factors? These existing studies focus on the 3.15 event, and the 3.27 event, which also has a huge impact, has not received sufficient attention. Furthermore, there are still several critical gaps for a better understanding of these two mega SDS events, such as, where are the source areas of these two SDS events? How extreme were the regional dust loadings during these two SDS events from a historical perspective? How does the enhanced dust aerosol loading affect the optical, microphysical, and radiative properties of aerosols? What are
the similarities and differences between the two events in terms of dust sources, aerosol optical, microphysical, and radiative properties, and meteorological drivers?

In this study, aerosol property data from the sun photometer observations in Beijing, combined with a variety of multiple satellite and ground-based observations as well as aerosol reanalysis data from the Modern-Era Retrospective Analysis for

Research and Applications, version 2 (MERRA-2) and meteorological reanalysis from the Fifth major global reanalysis produced by ECMWF (ERA5), were used to characterize the gigantic dust plume, reveal its sources and meteorological causes, and assess its impact on radiation balance. To be specific, this study will seek to (1) describe the three-dimensional vertical evolution features of the dust plumes during transport and assess its impact on ground visibility, (2) characterize the optical, microphysical and radiative properties of aerosols affected by dust plumes, (3) place the intensity of these two SDSs, with dust optical depth (DOD) as an indicator, in the context of the last two decades for NC and its sub-regions, (4) understand the circulation patterns that contribute to the formation of SDSs, and (5) explore the local meteorological anomalies that lead to changes in the condition of surface soil layers in dust source areas that favor enhanced dust emissions.

## 2 Data and methods

### 2.1 Satellite datasets

#### 2.1.1 Combined AOD from MODIS and VIIRS

Aerosol optical depth (AOD) is the column-integrated light extinction by aerosol particles. In this study, Level 2 daily AOD at 550 nm retrieved from MODIS on board the Terra and Aqua satellites and from Visible Infrared Imaging Radiometer Suite (VIIRS) on board the Suomi National Polar-Orbiting Partnership (SNPP) satellite during the 3.15 and 3.27 SDS events are used to examine the downstream propagation of the dust plume. Previous studies (Hsu et al. 2019; Sayer et al. 2019) have shown that MODIS and VIIRS AOD agree well with the Aerosol Robotic Network (AERONET) observations. We regridded Level 2 AOD products from the three sensors, which were derived using the Dark Target (DT) and Deep Blue (DB) algorithms, respectively, to a 0.1° × 0.1° grid and averaged among them to produce daily combined AOD. Note that only the AOD data flagged as good quality (quality flag = 2) or very good quality (quality flag = 3) was used.

#### 2.1.2. MODIS-retrieved long-term DOD time series

DOD, as one of the key parameters for characterizing the optical properties of dust aerosols, describing the columnar optical depth due to the extinction by mineral dust particles, has been widely used in dust-related studies (e.g., Pu and Ginoux, 2016; Song et al., 2021; Gkikas et al., 2021; Logothetis et al., 2021). The long-term DOD data in March for NC were derived from MODIS Level 2 aerosol property data retrieved using the DB algorithm (Sayer et al., 2019), which utilizes the radiance received by the blue channels to detect land aerosol loadings with bright surfaces (e.g., desert). In this study, MODIS DB Collection 6.1 aerosol products from both the Terra (MOD04_L2) and Aqua (MYD04_L2) platforms were used. Before retrieving DOD values, aerosol products including AOD at 550 nm, single-scattering albedo (SSA), and the Ångström exponent were first interpolated to a regular 0.1° × 0.1° grid using the nearest-neighbor algorithm. For AOD, only the AOD data flagged as good quality or very good quality was used, which effectively eliminated the influence of low-quality AOD on the DOD retrieval. Then, gridded DOD data were retrieved using these parameters as the input following the methods of

Pu and Ginoux (2018). In Pu and Ginoux (2018), the DOD over land was derived from MODIS DB aerosol products by using a continuous function relating the Ångström exponent to the fine-mode AOD established by Anderson et al. (2005) based on in-situ data. This approach, as summarized in Eq. (1), allows the separation of coarse-mode dust and fine particle contributions from the AOD:

$$DOD = AOD \times (0.98 - 0.5089\alpha + 0.0512\alpha^2) \tag{1}$$
$$(\omega < 1.0, \alpha < 0.3),$$

where $\alpha$ is the Ångström exponent and $\omega$ is the SSA. The DOD is obtained only when $\omega$ is less than 1.0 and $\alpha$ is less than 0.3 for dust owing to its predominance of coarse modes and its absorption of solar radiation.

To improve the spatial coverage of the limited DOD retrieval and also to incorporate inputs from both morning (~10:30 local time for Terra) and afternoon observations (~13:30 local time for Aqua), the combined daily DOD was derived by averaging the Terra and Aqua DODs when both sets of information were available, or by using either the Terra or Aqua DOD when only one set of information was available. Subsequently, these daily DOD data were averaged to monthly average DODs by requiring a minimum of three valid daily DOD retrievals in a month. This combined March DOD dataset is available from 2000 to 2021 (until 2003 from Terra, and thereafter a combination of both Terra and Aqua).

MODIS dust detection is subject to a number of uncertainties. Over land, the derived MODIS DOD here denotes the coarse-mode (aerodynamic diameters larger than 1 μm) contribution of dust only and does not include its fine-mode contribution. Estimates by Kok et al. (2017) suggest that the exclusion of submicron dust aerosol could induce around 3% underestimation of the global atmospheric dust mass load and around 15% underestimation of the global DOD. Terra and Aqua DOD values at daily and monthly scales have previously been validated with AERONET stations globally (Pu and Ginoux, 2018; Song et al., 2021). Spatially, when comparing the MODIS-derived DOD climatology with the DOD retrieved from the Cloud–Aerosol Lidar with Orthogonal Polarization (CALIOP) aboard CALIPSO satellite, the climatological mean of the MODIS DOD generally compares well with CALIOP (Pu and Ginoux, 2018; Song et al., 2021). Here we compare the combined daily MODIS DOD against AERONET stations in NC (Fig. S1). It should be noted that to date, there is no valid method to derive DOD from AERONET AOD measurements. Therefore, we use coarse-mode AOD (AODc) from AERONET measurements as a proxy for DOD (Pu and Ginoux, 2018; Song et al., 2021) to compare with our DOD datasets in March for NC.

For this purpose, we use the AERONET Version 3 spectral deconvolution algorithm (SDA) daily products (Level 2.0 data). Given that AERONET SDA only provide the AODc at 500 nm, the AERONET AODc is converted to 550 nm in this study using the Ångström exponent to compare with MODIS DOD retrievals. For March dust retrievals, between 2000 and 2021, there are 121 MODIS daily mean DOD retrievals collocated with 7 AERONET sites located within NC (Fig. S1). Results showed that the MODIS DOD is in good agreement with AERONET AODc (Person correlation coefficient = 0.82), although the former generally overestimated latter in NC (root-mean-square error = 0.28).

### 2.1.3. CALIOP dust extinction profiles

To characterize the vertical profile of the dust plume, the Level 2 daily 532 nm aerosol profile product (05kmAPro, V4.21) that contains aerosol depolarization, backscatter, and extinction profile from CALIOP/CALIPSO (Winker et al., 2010) was used. We did not directly use the extinction profile of the "dust" or "polluted dust" type as determined by the CALIPSO aerosol type classification algorithm. Using "dust" alone would result in an underestimation of the actual atmospheric dust loading, while introducing the sum of "dust" and "polluted dust" would overestimate the dust loading (Han et al., 2022). Therefore, we use the methodology in Yu et al. (2015) to derive the dust extinction profile. To reduce uncertainty, only high-quality extinction profile data with a CAD (cloud aerosol discrimination) score of between −100 and −90 were used. The aerosol profile product also provides an extinction quality control flag (Ext_QC) to indicate problematic retrievals. This study only uses layers with Ext_QC values of 0, 1, 18, and 16 (Winker et al., 2013).

For each aerosol backscatter coefficient profile, we infer the ratio of dust to total backscatter ($f_d$) at each altitude from the following equation:

$$f_d = \frac{(\delta - \delta_{nd})(1 + \delta_d)}{(\delta_d - \delta_{nd})(1 + \delta)} \tag{2}$$

where $\delta$ is CALIOP observed particulate depolarization ratio, $\delta_d$ and $\delta_{nd}$ are a priori knowledge of depolarization ratios of dust and non-dust aerosols respectively. To account for various types of non-dust aerosols with different depolarization ratio and for the variability of dust shape and size, we follow Song et al. (2021) and use the $f_d$ that was based on the mean of the lowest ($\delta_d = 0.30$ and $\delta_{nd} = 0.07$) and the highest ($\delta_d = 0.20$ and $\delta_{nd} = 0.02$) dust scenario. By assuming a dust lidar ratio (LR) (i.e., extinction-to-backscatter ratio) of 40 sr at 532 nm (Yu et al., 2015), we derive dust extinction coefficient (DEC) profile from dust backscatter coefficient.

### 2.1.4. Himawari-8 dust RGB composite imagery

Himawari-8 geostationary satellite operated by Japan Meteorology Agency (JMA) provides images of East Asia and Western Pacific Region at a frequency of every 10 min, day and night (Bessho et al., 2016). This allows for monitoring the source, genesis and movement of dust at high temporal resolution. Two brightness temperature (BT) differences (between 12.4 and 10.4 µm and between 10.4 and 8.6 µm) and the BT at 10.4 µm are rendered to red–green–blue (RGB) beams to highlight the presence of Aeolian dust (Shimizu, 2020). Note that although the high temporal resolution of Himawari-8 combined with the RGB composite images offers the potential to identify specific features forward or backward in time, the ability of the dust RGB imagery to identify dust, via its characteristic magenta coloring, is strongly dependent on the column water vapor, the lower tropospheric lapse rate, and dust altitude (Brindley et al., 2012). In this study, to highlight the dust phenomenon, we use gamma value correction to enhance high/low brightness intensity pixel values (BYTE). The formula for such correction is

$$BYTE = \left[\frac{BT - M}{MAX - MIN}\right]^{\frac{1}{\Gamma}} \tag{3}$$

where BT is brightness temperature differences (i.e., red and green beams) or BT for blue beam, MIN and MAX represent the ranges of BT differences/BT, and $\Gamma$ is the gamma value. Here, we use recommended dust RGB composite thresholds for Himawari-8 (Shimizu, 2020): MIN (MAX) is −4.0 (2.0), 0.0 (15.0), 261.0 (289.0) for red, green and blue beams, respectively, and $\Gamma$ is 1.0, 2.5 and 1.0, respectively. In this study, we use Himawari-8 dust RGB composite imagery to illustrate the source, genesis and movement of dust plumes during the two SDS events.

## 2.2. Observation datasets

### 2.2.1. Sun photometer

A Cimel CE-318 sun photometer installed on the roof of the Chinese Academy of Meteorological Sciences (CAMS), Beijing, China (39.93°N, 116.32°E; 106 m), has been in operation since 2012. This sun photometer, which is named "Beijing-CAMS", operates in both the China Aerosol Remote Sensing Network (CARSNET) and AERONET and is calibrated at Izaña, Tenerife, Spain, together with the AERONET program (Che et al., 2009). Beijing-CAMS is the main site of CARSNET, but measurements are also uploaded to the AERONET data archive. The measurements of aerosol optical, microphysical and radiative properties at the Beijing-CAMS site can provide a reference for assessing several aspects of the dust aerosol loading intensity, evolution and its impact during the downstream transport of the dust plume.

In this study, the cloud-screened instantaneous AOD data at multiple wavelengths were calculated by using the ASTPwin software (Cimel Electronique), and extinction Ångström exponents (EAE) were calculated from the AODs for wavelengths of 440 and 870nm. The fine-mode fraction (FMF) is described as the fraction of fine-mode particles of total $AOD_{440nm}$. To perform a spatio-temporal synergistic analysis between satellite and ground-based observations, the AOD measurements in two adjacent channels (i.e., 440 and 675nm) from sun photometer was interpolated to 550nm for satellite retrieval, using a second-order polynomial fit to ln (AOD) vs. ln (wavelength) (Eck et al., 1999).

The aerosol microphysical properties, including volume size distributions ($dV(r)/d\ln r$) in 22 size bins for particle radii 0.05–15 μm; the total, coarse-mode, and fine-mode volume concentrations ($Volume_t$, $Volume_c$, and $Volume_f$, respectively); the total, coarse-mode, and fine-mode aerosol effective radii ($R_{eff_t}$, $R_{eff_c}$, and $R_{eff_f}$, respectively); the absorption AOD (AAOD); SSA; and the absorption Angström exponent (AAE), were retrieved from the almucantar sky irradiance measurements in conjunction with measured spectral AOD at 440, 670, 870, and 1020nm using the algorithms of Dubovik et al. (2002, 2006).

The shortwave direct aerosol radiative forcing (DARF in W m$^{-2}$) was calculated by the radiative transfer module of AERONET under the assumption of cloud-free conditions (García et al., 2008, 2012). The DARF at the top of the atmosphere (TOA) and the Earth's surface (bottom of the atmosphere, BOA) was defined as the difference in the shortwave radiative fluxes with and without aerosol effects in Eqs. (4) and (5) as follows:

$$DARF_{TOA} = F_{TOA}^{\uparrow 0} - F_{TOA}^{\uparrow} \tag{4}$$

$$DARF_{BOA} = F_{BOA}^{\downarrow} - F_{BOA}^{\downarrow 0} \tag{5}$$

where $F_{TOA}^{\uparrow 0}$ and $F_{BOA}^{\downarrow 0}$ denote the broadband fluxes with no aerosols at TOA and BOA, respectively. DARF$_{TOA}$ is the reflection of solar radiation by aerosols back to space, while DARF$_{BOA}$ indicates the combined effects of absorption and scattering of solar radiation by aerosols. The findings of García et al. (2008) show that the error for the observed solar radiation at the surface on a global scale was +2.1 ± 3.0% for an overestimation of about +9 ± 12 W m$^{-2}$. Subsequently, the DARF at atmosphere (DARF$_{ATM}$) was defined as the difference between DARF$_{BOA}$ and DARE$_{TOA}$ as follow:

$$DARF_{ATM} = DARF_{TOA} - DARF_{BOA} \tag{6}$$

Defined this way, a negative value for DARF indicates aerosol cooling effects, while positive values imply warming.

### 2.2.2. Horizontal visibility

Horizontal visibility is closely related to air quality and can be an important indicator of the quality of the atmospheric environment in most scenarios (Gui et al., 2020). In this study, the hourly visibility observations from ~1600 national surface meteorological observation stations across NC provided from the China Meteorological Administration (CMA) during the two SDS events were used to characterize the impacts of the dust plume on air quality. In order to minimize the effect of relative humidity (RH) on horizontal visibility, RH values > 40 and < 99% were converted to the equivalent visibility in dry conditions (i.e., RH < 40%). The correction formula is expressed as VIS/VIS(dry) = 0.26 + 0.4285 $\log_{10}$ (100 – RH) (Rosenfeld et al., 2007). The hourly visibility data accompanied by the presence of fog or precipitation were excluded from this study.

### 2.2.3. Ground-based lidar observations from AD-Net

Asian Dust and aerosol lidar observation Network (AD-Net) (Shimizu et al., 2004; Sugimoto et al., 2008) is a lidar network for continuous monitoring of the vertical profile of dust and other aerosols (e.g., anthropogenic aerosols, biomass burning aerosols, and volcanic ash aerosols), with the aim of investigating the implications of aerosols on climate, environment and health in East Asia. AD-Net is composed of a dual-wavelength (1064 nm, 532 nm) polarization-sensitive Mie-scattering lidar (Sugimoto et al., 2008), which has been distributed to more than 20 sites in East Asia. In this study, the DEC profile at 532 nm provided by an AD-Net site named "Zamynuud" were used to explore the impacts of long-range dust transport on downstream areas. The geographical location of the "Zamynuud" site is shown in Fig. 1. Each lidar from AD-Net takes a 5-minute measurement every 15 minutes, generating an observation data file with the vertical resolution of 30m. For AD-Net, the Klett's inversion method was employed to derived the extinction coefficient, after applying a geometrical-form-factor correction (Sugimoto et al., 2003). With the assumption of external mixing of dust and spherical aerosols, the ratio ($R$) of contribution of dust in the extinction coefficient is calculated as follows:

$$R = \frac{(\delta_a - \delta_2)(1 + \delta_1)}{(1 + \delta_a)(\delta_1 - \delta_2)} \tag{7}$$

where $\delta_a$ is the observed aerosol depolarization ratio (ADR). $\delta_1$ and $\delta_2$ are ADRs of dust and air-pollution aerosols. The values of $\delta_1$ and $\delta_2$ were determined empirically and are 0.35 and 0.05, respectively (Sugimoto et al., 2003; Shimizu et al., 2004).

### 2.3. Reanalysis datasets

### 2.3.1. MERRA-2 aerosol reanalysis

MERRA-2 products from the NASA's GMAO (Global Modeling and Assimilation Office) were used to identify the sources of the two SDS events. The MERRA-2 reanalysis was generated using the Goddard Earth Observation System (GEOS-5) with a 3D variational data assimilation system that assimilates a large number of observational datasets (Buchard et al., 2017). The MERRA-2 hourly dust emissions at a 0.5° ×0.625° horizontal resolution during these two SDS events were used. GEOS-5, based on Ginoux et al. (2001), simulated dust emissions, which were resolved in five size bins with diameter bounds at 0.1, 1.0, 1.8, 3.0, 6.0, and 10.0 μm, respectively. By comparing with satellite and ground-based observations, Yao et al., (2020) demonstrated the ability of MERRA-2 in characterizing the three-dimensional evolution of dust aerosols during an extreme SDS event in East Asia.

### 2.3.2. ERA5 atmospheric reanalysis

ERA5 reanalysis data (Hersbach et al., 2020), at a 0.25° × 0.25° horizontal resolution, for March 2000–2021, were used to investigate both atmospheric circulation extremes and the local meteorological anomalies associated with two strong SDSs in March 2021. In this study, both daily and monthly variables were used, the former including daily wind fields at 10 m and 700 hPa and daily sea level pressure (SLP), geopotential height (GH) and temperature fields at 700 hPa, which were used to explore the atmospheric conditions driving the emission, transport and deposition processes of the two extreme SDSs; and the latter including temperature at 2 m, snow depth (SD), PPT and VSW at 0–7 cm depth, which were used to resolve the historical meteorological anomalies driving the March 2021 dust anomaly. As a complement, the hourly SLP and wind field on March 14 were used to analyze the atmospheric circulation background for the day before the outbreak of the 3.15 event.

## 3. Results and discussion

### 3.1. Overview of the two severe SDS events in March 2021

Several studies have been performed to reveal the transport processes of dust aerosols during the 3.15 event and their impacts on near-surface air quality using $PM_{10}$ concentration as indicator (Liang et al., 2021; Filonchyk, 2022; Filonchyk and Peterson, 2022). However, no studies have focused on the 3.27 event and the differences between the two events have not been explored. This section will provide an overview of these two SDS events based on satellite RGB images, horizontal visibility observations, and multi-satellite fusion to reveal the similarities and differences between them from different

perspectives. Here we show a sequence of Himawari-8 dust RGB composite still images at 13:00 CST during the 3.15 and 3.27 SDS events to illustrate the day-to-day evolution of the dust plumes (Fig. 2). Overall, the transport pathway of the dust plume during the two SDS events was mainly controlled by the movement trajectory of a cyclone (namely, the Mongolian cyclone). On March 15, a powerful cyclone (maximum wind speed more than 15 m s$^{-1}$ at 10m) located in northeastern China (NEC) uplifted dust aerosols emitted from the dust source (i.e., the GD, which will be discussed in detail in Section 3.4) into the atmosphere, forming a dust belt. From March 16 to March 17, the dust RGB images successfully captured the dust plumes has been carried to southern Hebei, Shandong, and northern Henan. Moreover, the continued emissions from the GD directly contributed to the formation of dust hotspots in southern Mongolia and western Inner Mongolia. Subsequently, these dust aerosols began to transport downstream driven by northwesterly winds and impacted regions such as the North China Plain (NCP) and north-central China again.

During the 3.27 event, a cyclone originating in eastern Mongolia picked up the dust aerosols from the dust source and transported them to the NCP, Liaoning, Jilin, Heilongjiang, and the northwestern Sea of Japan over the next two days as the cyclone developed and strengthened and moved eastwards. Notably, it was found that, although the dust plume near the NCP on March 15 captured by Himawari-8 was morphologically similar to the dust plume on March 28, the latter was located slightly to the south. The differences in the spatial location of the dust plume were mainly caused by the differences in the intensity, extent and location of the cyclones on these two days (for detailed information please see Section 3.5). Overall, despite the relatively short duration of the 3.27 event, the extent of its impact on the NCP cannot be ignored.

Fig. 3 shows the daily mean horizontal visibility maps for the 3.15 and 3.27 events, respectively. Note that the horizontal visibility here has been corrected by the RH threshold (see Section 2.2.2). The RH-corrected visibility filters out the effects from high RH events and instead highlights the effect of dust in weakening visibility (see Fig. 3 and Fig. S2). CMA station records show that, during the 3.15 event, horizontal visibility first reached a minimum on March 15 in most of NC, including Gansu, southwestern Inner Mongolia, Ningxia, northern Shaanxi, northern Shanxi, Hebei and Beijing, with the number of stations with daily mean visibility below 500 m reaching 19, due to dust plume deposition. Such remarkable contribution of dust aerosols to air quality is also supported by the results revealed by Filonchyk (2022) using PM$_{10}$ observations. Influenced by SDS, PM$_{10}$ concentrations in some regions of China were found to exceed 7000.0 μg m$^{-3}$ on March 15. Similarly, Liang et al. (2021) claimed that the 3.15 SDS event was the most severe in China in the past decade, with PM$_{10}$ concentration in Beijing reaching 6450 μg m$^{-3}$. Our results show that on March 15, the instantaneous surface horizontal visibility was below 50m near the lower limit of the monitoring threshold at seven sites (Fig. S3). On March 16, a large amount of dust aerosol was further transported downstream, and its impact spread to Shandong, Henan, and the northern part of Jiangsu and Anhui, with the number of stations with daily mean visibility below 500m (1.0km) reaching 4 (17). On March 17, under the influence of southeasterly winds blowing from the Yellow Sea (see Fig. 2c), the dust plume stopped continuing southwards and began to reflux and gradually deposit, and the intensity of its influence weakened significantly compared with the previous two days. Nevertheless, dust concentrations remained at extremely serious pollution levels in the areas near the dust source, such as Gansu, Ningxia, and southwestern Inner Mongolia, which directly

contributed to the low-level visibility hotspots in the aforementioned regions. On March 18 and 19, the daily mean visibility gradually increased to above 9 km in the downstream area of the dust source except in Gansu, Inner Mongolia and Qinghai. The dust plume eventually began to dissipate on March 20 under the action of strong northwesterly winds (Fig. 2f), but the dust plume passing through the NCP area once again worsened local air quality, as visibility was only 3–6 km at most sites (Fig. 3f).

A week after the end of the 3.15 event, a new dust plume swept across most of the north again. Overall, the 3.27 event was weaker than the 3.15 event in terms of magnitude, scope and duration of the dust impact. Specifically, the 3.27 event started on 27 March, reached its peak on 28 March, and then gradually dissipated on 29 March. The inconsistency in satellite imagery and visibility in NCP on March 27 (Figs. 2g and 3g) are mainly attributed to low visibility event caused by excessive fine particulate matter emissions from local anthropogenic activities. Despite its relatively short duration, the impact of the 3.27 event (especially on March 28) still covered areas including southern Inner Mongolia, Hebei, Shanxi, and Shandong. Visibility observation records show that the daily mean (instantaneous) visibility less than 1.5 (0.5 km) at 124 (19) sites.

Satellite retrievals not only provide a broader perspective than ground-based observations, but also allow capturing the dynamic evolution of the optical properties of dust aerosols during transport. Fig. 4 shows the MODIS and VIIRS combined daily mean AOD maps during the 3.15 and 3.27 events at a frequency of every other day. The accuracy of the MODIS and VIIRS instantaneous AOD values retrieved using individual algorithms (i.e., DT and DB algorithms) was confirmed by a synergistic comparison with ground-based sun photometer observations located in Beijing (i.e., Beijing-CAMS site) (Fig. 5). To complete the spatial collocation, the satellite-retrieved AOD values is represented by the average value of all the satellite product pixels within a 25 km radius (Sayer et al., 2013) around the Beijing-CAMS site. The results show that both the MODIS and VIIRS instantaneous AOD values are in high agreement with ground-based observations. Here, multi-satellite fusion provides more available retrievals and regional details by incorporating the observed dust loading at different satellite transit times than the individual satellite data sources used in the previous similarity studies (e.g., Filonchyk, 2022; Filonchyk and Peterson, 2022). In addition, given that AOD is an aggregate information for all aerosol types, we further give the MODIS-retrieved DOD to quantify the magnitude of the two dust transport processes (Fig. S4), while the ratio of DOD to AOD (Fig. S5) elucidates the contribution of dust particles in the aerosol plume. Overall, the evolutionary pattern of AOD or DOD is consistent with that of visibility observations (see Fig. 3). On the first day of the 3.15 event (March 15), it is clear from these maps that there were dust plumes as wide as 2500 km (confined within 40–50°N), crossing most of NC in a meandering path. Due to the large amount of mineral dust aerosols emitted into the atmosphere, a resultant peak AOD (DOD) of about 4.0 (3.0) occurred on March 15–17. From March 18 to March 20, the dust loading in the area near the source and downstream gradually decreased as the deposition and emission of dust weakened. Early in the 3.27 event (March 27), MODIS captured an enhanced polluted hotspot (AOD > 2.0), which was located in southern Mongolia. This suggests that the 3.27 event was associated with intensified emissions in this region. On March 28, dust plumes carrying large amounts of dust aerosols were transported to the North China Plain (NCP) and Liaoning under the action of atmospheric circulation, with the

AOD reaching the range of 3–4. Such a strong dust column loading observed on 3.28 is comparable in magnitude to the enhanced dust plume experienced on 3.15. Notably, a low DOD value (< 0.5) was observed at the dust source on March 28 (Fig. S4), which implies that the intensity of dust emissions was significantly lower on March 28 than on March 27.

### 3.2. Aerosol optical and microphysical properties

In this study, the dynamical evolution of the columnar aerosol optical, microphysical, and radiative properties during the two SDS events were obtained using the continuous ground-based sun photometer observations at the Beijing-CAMS site. Fig. 5 shows the daily variation of AOD, EAE, and FMF. Influenced by the dust plume, the AOD in Beijing increased significantly during the period from 13 to 17 h (CST) on March 16, reaching a maximum value of about 2.0 on March 17, which was about 10 times higher than the AOD (i.e., 0.2) under clean condition in the morning of March 16. Temporally, the enhanced AOD in Beijing lasted for nearly one day. With the end of the 3.15 SDS event, the AOD decreased to 0.1 on March 20. From 15 h on March 16 until 09 h on March 17, EAE values between 0.1 and 0.3 and FMF values between 0.2 and 0.3 together reveal that the enhanced aerosol loading is dominated by coarse-mode particles, with a smaller contribution from fine-mode particles.

On the second day of the 3.27 event (i.e., March 28), although the duration of the enhanced AOD was shorter than that on March 17, its peak was higher (~2.5). During the high aerosol loading period, the observed EAE values are usually less than 0.1 (or even less than 0 in some cases) and the extremely low FMF values (< 0.2) suggest a dominant role for coarse-mode aerosol particles with particle sizes even higher than on March 17.

The significant contribution of coarse-mode particles during the two high aerosol loading events was also confirmed from the aerosol volume-size distributions (Fig. 6). Although the dust aerosols on both March 17 and March 28 consisted mainly of coarse particles clustered at a radius of about 2.2 μm, the volume peak of the latter ($dV(r)/dlnr = 2.24$ $\mu m^3\,\mu m^{-2}$) was three times higher than that of the former (0.78 $\mu m^3\,\mu m^{-2}$). The extreme volume concentrations observed during the 3.27 event were almost twice as high as those observed (~1.2 $\mu m^3\,\mu m^{-2}$) during a strong SDS event that occurred in May 2017, which was considered as the worst in the last five years (Filonchyk et al., 2021). In terms of aerosol effective radii and volume concentrations (Table 1), the effective radius of coarse-mode particle was about 1.661 ± 0.058 μm on March 17, and the volume concentration increased significantly, from 0.121 ± 0.007 $\mu m^3\,\mu m^{-2}$ on March 16 to 1.022 ± 0.056 $\mu m^3\,\mu m^{-2}$ on March 17. Throughout the observation period, the highest volume concentration in coarse-mode particles occurred on March 28 at about 2.007 $\mu m^3\,\mu m^{-2}$, followed by an increase in the effective radius to 1.850 μm. These results are consistent with our previous inference from AE and FMF that the Beijing area experienced an intrusion of coarse-mode dust particles with larger effective radii and volume concentrations during the 3.27 event than during the 3.15 event.

The dynamic evolution of the dust plume in the vertical direction during these two SDS events was characterized using continuous ground-based Lidar observations. Fig. 7 shows the vertical evolution of the DEC at 532nm for the AD-Net site named "Zamynuud" located at the edge of Inner Mongolia. Influenced by the 3.15 mega SDS event, the dust plume was first

transported to the near-surface layer from 6 km altitude at 04:00 to 18:00 CST on March 16, and formed an intense aerosol layer with a DEC larger than 0.5 km$^{-1}$ at an altitude of about 1.5 km. Subsequently, in the altitude range of 0–2 km, the thickness of the aerosol layer with a DEC of $\sim 0.05-0.2$ km$^{-1}$ continuously expanded to about 1 km and remained in a stable

phase for more than 12 hours. At 18:00 CST March 17, as the near-surface aerosols dissipated, the two dust plumes were once again transported to over north-central Inner Mongolia through two different altitude pathways (starting at 2 km and 4 km altitude, respectively). By this time and until 18:00 CST March 18, the dust plume located in the lower layers was transported to the near-surface, which enhanced the DEC to 0.2 km$^{-1}$ and formed a dust aerosol layer with a thickness of ~2 km. In contrast, the strong dust plume (DEC > 1.0 km$^{-1}$) located in the upper layers carrying a large amount of dust aerosols

was maintained at an altitude of 1−3 km for nearly 1 day before being transported far from the observation site at 18:00 CST March 19. After this, the high-altitude dust plume gradually dissipated as the dust aerosols diffused and were partially deposited, and the observation site was only intermittently affected by surrounding dust transport.

Similar to the 3.15 event, north-central Inner Mongolia was also influenced by the transport of dust plumes during the 3.27 event. Specifically, two moderate dust plumes (DEC~0.05 km$^{-1}$) had affected the lower (0–2km) and upper layers (3–

395 4km) in north-central Inner Mongolia, respectively, on March 26. Subsequently, the observation site may have been affected by a combination of lower-layer and high-altitude transport. Clearly, the dust plume with a DEC of about 0.2 km$^{-1}$ at an altitude of about 3.0 km started to transport downward since 08:00 CST March 27 and reached the near-surface 5 hours later. This transport process allowed for a rapid accumulation of near-surface dust aerosols, resulting in the surface DEC increasing to about 1.0 km$^{-1}$.

As a supplement to ground-based Lidar observations, space-based CALIOP/CALIPSO observations were also utilized in this study to better characterize the vertical evolutionary structure of the dust plume in space over time. Fig. 8 displays the CALIOP-derived DEC at 532nm over north-central China (March 15 and 27), the region spanning the Yellow Sea and NEC (March 16), and the region spanning northern Beijing, the Bohai Bay, and the eastern coast (March 28). On March 15, a large amounts of dust aerosols forming an enhanced dust layer within 2–6 km between 36°N and 41°N, where dust with

DEC > 1.0 km$^{-1}$ was located in the lower layer (2–3 km). Notably, we found that the CALIOP-derived DEC profiles appear to be missing throughout the entire height range between near-surface and 2 km in the dust source area. Most studies show that the official cloud-aerosol discrimination algorithm are usually able to correctly identify dust aerosols (Omar et al., 2009; Kim et al., 2018), but dust aerosols may be misclassified as clouds when severe dust storms are encountered (Han et al., 2022). So, when the dust concentration is extremely high and the DOD is over 3.0, it is difficult to obtain an accurate and

complete dust profile because the attenuation signal received by CALIOP may be biased beneath the thick dust layer (Han et al., 2022; Pu and Jin, 2021). On the following day, March 16, the dust plume was continuously transported downstream and lifted to an altitude of more than 8 km by the interaction of wind and topography, but its strength was largely reduced with a DEC of about 0.01 to 0.1 km$^{-1}$.

In contrast, the lifting altitude of the dust plume on March 27 was lower than that on March 15, with the top of the dust

plume at about 5 km. The difference in lifting altitude of the dust plume between the two days was also confirmed by

ground-based lidar observations near the two CALIOP/CALIPSO tracks (Fig. 7). Nevertheless, CALIOP detected a dust layer spanning 30°N to 45°N on March 27, with an enhanced DEC ($> 1.0$ km$^{-1}$) located mainly between 39° and 45° (i.e., near the GD). In terms of intensity, the thick near-surface dust layer observed on March 27 was comparable to that of March 15, which led to the unavailability of the DEC profiles retrieved from CALIPSO at some locations. After 1 day of transport (March 28), these dust plumes were transported to Bohai Bay and the eastern coast and were mainly constrained to the altitude range of 0–3 km between 36° and 40°N. The substantial accumulation of dust aerosols (DEC $> 1.0$ km$^{-1}$) in the lower atmosphere directly led to a significant deterioration of the near-surface visibility (see Fig. 3h), which is also confirmed by the Himawari-8 image (Figs. 1b and 2h) and the MODIS and VIIRS combined AOD (Fig. 4h).

### 3.3. Direct aerosol radiative forcing

The DARF is one of the important parameters for the assessment of aerosol radiative effect on climate. Fig. 9 demonstrates daily mean variations of the shortwave DARF$_{BOA}$, DARF$_{TOA}$, and DARF$_{ATM}$ during the two SDS event. During the 3.15 event, the BOA in Beijing fluctuated significantly, with the DARF$_{BOA}$ increased from $-33.4 \pm 0.4$ W m$^{-2}$ on March 16 to $-218.1 \pm 14.4$ W m$^{-2}$ on March 17. The increase in DARF$_{BOA}$ is attributed to the enhanced aerosol loading (AOD of about 2.0) during the pollution episode leading to a decrease in solar radiation reaching the ground, causing an enhancement of the cooling effect. Meanwhile, the enhanced absorption capacity (SSA = 0.915) associated with increased dust loading led to the weakened scattering of atmospheric aerosols (Table 1), and part of the radiation is absorbed by the dust aerosols, causing an increase in the cooling effect at the TOA, which eventually led to an increasing trend of DARF$_{TOA}$ as well, from $-23.9 \pm 1.1$ W m$^{-2}$ on March 16 to $-115.9 \pm 15.1$ W m$^{-2}$ on March 17. The presence of large amounts of dust aerosols with weak absorption also directly contributed to the increase in atmospheric radiative heating, with DARF$_{ATM}$ on March 17 increased by $+ 92.6$ W m$^{-2}$ relative to non-dust day (March 16).

During the 3.27 event, we found that DARF$_{BOA}$ was higher on March 28 than on March 17. The higher DARF$_{BOA}$ ($-317.7$ W m$^{-2}$) on March 28 was associated with higher aerosol loading (AOD of about 2.5), which directly led to a stronger surface cooling effect than on March 17. The weakening of the aerosol scattering (SSA = 0.900) due to the enhanced instantaneous dust loading on March 28 contributed to a stronger cooling effect at the TOA on March 28 than on March 17, reaching $-127.1$ W m$^{-2}$. Moreover, the presence of large amounts of coarse-mode dust particles with high volume concentrations (2.007 μm$^3$ μm$^{-2}$) on March 28 increased the potential for instantaneous absorption of radiation by the atmosphere, accounting for the higher DARF$_{ATM}$ on March 28 ($+190.6$ W m$^{-2}$) than on March 17. Generally, the instantaneous DARF on March 28 estimated in this study was stronger than similar studies previously performed during several strong SDS events, such as in Beijing in May 2017 (Filonchyk et al., 2021) and over the Indo-Gangetic Basin in May 2018 (Tiwari et al., 2019).

### 3.4. Dust source region and emission

Currently, models still face huge challenges in accurately quantifying dust emissions, due to limitations such as uncertainties in the dust source locations and dust emission parameterization schemes (Kok et al., 2020). Aerosol reanalysis involving the assimilation of a large number of observations is considered a valuable tool for evaluating dust processes in climate models (Wu et al., 2020; Zhao et al., 2021). Admittedly, aerosol reanalysis also carries some uncertainties; however, it was still expected to provide a valuable reference for identifying the sources of these two dust processes. Dust emissions during these two SDS events were characterized by MERRA-2 aerosol reanalysis data. Fig. 10 displays the MERRA-2 daily mean dust emissions for all size bins during the 3.15 and 3.27 events. Overall, in both SDS events, MERRA-2 identified two sources of enhanced dust emissions: one in the TD and the other in the GD across southern Mongolia and northwestern Inner Mongolia. Although dust emissions from the TD were more intense than those from the GD on most days, dust particles from the TD were not susceptible to be transported outside the Tarim Basin owing to the prevailing surface easterly winds (see wind fields in Fig. 2). Typically, the TD being surrounded by mountains on three sides, and the surface wind being dominated by an easterly with low speeds at high altitudes in spring, results in most spring dust over the TD being re-deposited after uplift (Chen et al., 2017b). Therefore, the source of these two SDS events can be basically determined as the GD. In the following, the focus is mainly on the variability of dust emissions in the GD.

As shown in Fig. 10, on March 15, MERRA-2 captured intensified dust emissions in the GD within Inner Mongolia, with a daily mean dust emission level of 5–20 $\mu g\,m^{-2}\,s^{-1}$ and a daily maximum hourly dust emission level of more than 50 $\mu g\,m^{-2}\,s^{-1}$ (Fig. S6). Such a level of dust emission intensity was bound to have been largely responsible for the deteriorating visibility in most areas of NC. It is worth noting that on March 15 the dust emissions from the GD located in northwestern Inner Mongolia, China were significantly higher than the intensity of dust emissions from the GD located in southern Mongolia, which raises the question as to whether the GD in China was the original source of the 3.15 SDS event. Fig. 11a further illustrates the MERRA-2 dust emissions and surface wind field for the day before the 3.15 event (March 14). The results show that the 3.15 SDS event was in fact first triggered by enhanced dust emissions from the GD in southern Mongolia, followed by the transport of large amounts of blowing dust aerosols into China by northerly winds that were enhanced by mixing with locally emitted dust aerosols. This suggests that, apart from the dust source areas in China, changes in surface conditions in the dust source areas of neighboring countries also need to be given more attention to reduce the frequency and effects of SDSs, especially in the context of climate change (Wu et al., 2021). Moreover, the dynamics of dust emissions are mainly regulated by the synoptic systems. The 3-h dust RGB imageries on March 14 clearly show the interaction between the synoptic systems and dust emissions, which is not always evident in the still images (Fig. S7). The dust plume formed at 12:00 CST on March 14 is triggered by the strong wind associated with the movement of the convective system. These findings are consistent with the results of Jin et al. (2022) using inverse modelling. They revealed that wind-blown dust emissions originated from both China and Mongolia contribute to the SDS events that occur in spring 2021. In the next 2 days (March 16 and 17), dust from the GD continued to be emitted into the atmosphere, but its intensity

tended to weaken. From March 18 to 20, although the emission of dust was enhanced compared with the previous two days, the uplifted dust aerosols were usually confined to the local area or lifted to higher altitudes, and did not have much impact on the near-surface in the downstream region.

On March 27, MERRA-2 identified that the second dust process originated from a dust source similar to the first process but in a slightly more easterly location. Although the 3-h dust RGB imagery has identified dust activity in south-central Mongolia associated with the incipient Mongolian cyclone at 18:00 CST on March 26 (Fig. S8), the enhancement of dust emissions is mainly controlled by the development and movement of the cyclone on March 27 (Fig. S9). In terms of the dust emission intensity (Fig. 10 and Fig. S6), although the emission intensity from dust sources within Inner Mongolia during the second process was lower than that of the first process, the intensity within Mongolia was significantly higher than that of the first process. Combining the wind fields (Fig. 2g), satellite images and dust emissions diagnosed by MERRA-2, it can be inferred that the 3.27 event initially originated from the dust source area in southeastern Mongolia. In the following two days (March 28 and 29), a significant reduction in the intensity of dust emissions was observed.

### 3.5. Atmospheric circulation patterns for controlling the emission and transport of dust

Most of the SDSs in NC have long been typically related to Mongolian cyclones and associated near-surface gales behind them (Zhu et al., 2008). The Mongolian cyclone, also known as the Mongolian low pressure, occurs or develops in Mongolia and is often accompanied by fronts, which mostly occur on the central and eastern plateaus of Mongolia on the leeward slopes of the terrain. The variation in the location and intensity of Mongolian cyclones will affect how dust is transported across NC. Here, the ERA5 meteorology associated with the 3.15 event is firstly analyzed by focusing on the SLP, temperature, and wind vectors. Fig. 11b displays the spatial patterns of the daily mean SLP, temperature at 2m, and wind vectors at 10 m on the day before the 3.15 event (March 14). On this day, an exceptionally strong Mongolian cyclone developed on the east side of Mongolia, accompanied by an extremely dense pressure gradient difference between the cyclone and the cold high pressure center on the west side, with the difference in daily average SLP reaching ~50 hPa (Fig. 11b). Induced by such a large pressure gradient difference, northerly gusts exceeding 20 m s$^{-1}$ ensued at the surface near Mongolia (Fig. S10), contributing to the enhancement of dust emissions in Mongolia and driving dust transport to the southeast. This triggering mechanism was also confirmed by Filonchyk (2022). In terms of timing, such a strong near-surface northerly wind was formed mainly from the afternoon of March 14 (~13:00 to 18:00 CST) (Figs. S7 and S10), and it was from that time that a large amount of dust aerosols blown by the gales in Mongolia began to be rapidly transported to Inner Mongolia, China. Next, by analyzing the ERA5 meteorological fields, including the GH, temperature and wind vectors at 700 hPa and the SLP, the focus switches to elucidating how the movement of the Mongolian cyclone during dust episodes controls the dust transport process.

Fig. 12 displays the evolving spatial patterns of the daily mean GH, temperature, and wind vectors at 700 hPa from March 15 to 20 and from March 27 to 29. The evolution of the daily mean SLP was shown in Fig. S11. In general, both SDS events were triggered by an exceptionally strong Mongolian cyclone generated at nearly the same location (along the central

and eastern plateau of Inner Mongolia) in conjunction with the surface-level cold high-pressure system at the rear (Fig. S11), albeit with differences in magnitude and spatial extent of impact. Specifically, on March 15, the Mongolian cyclone was located at the center of (49°N, 125°E) with a maximum height of ~2740 gpm. The northerly winds on the west side of the

515 Mongolian cyclone combined with the northwesterly airflow at high levels will have led to the cold air at the back of the cyclone (west side) moving southwards to create near-surface cooling. Meanwhile, the strong atmospheric pressure difference generated between the cold high pressure (representing cold air) and the Mongolian cyclone (low pressure system) produced windy weather at the back of the cyclone. This configuration of synoptic systems provides favorable dynamic conditions for dust emissions from the GD and transported along the direction of cyclone movement, which directly led to

520 visibility reaching the minimum of this process on March 15. This synoptic system has been broken since March 16 with the eastward movement of the Mongolian cyclone, resulting in weakened near-surface cold high pressure and near-surface winds (see Fig. S11). By March 17–20, the Mongolian cyclone had weakened further and drifted southeastwards; and meanwhile the northwesterly wind behind the trough continued to drive the dust plume eastwards, affecting most of NC and the southeast coast.

On March 27, a strong Mongolian cyclone regenerated in the southeastern part of Mongolia along the border with China, with a maximum height of about 2790 gpm. Such a strong cyclone caused a rapid decrease in near-surface pressure, with the lowest value of SLP being about 990 hPa (Fig. S11). Under the control of such a strong low-pressure system, northwesterly gusts exceeding 15 m s$^{-1}$ ensued at the surface near eastern Mongolia. Although the difference in GH between the two cyclonic centers was only 50 gpm, the Mongolian cyclone that developed on March 27 was weaker than the

Mongolian cyclone that developed on March 15 in terms of overall intensity and the cold high-pressure system at the rear (Fig. S11g), which explains the difference in the amount of dust emissions during these two SDS events. In addition, although we observed a more westerly cyclone position on March 27 than on March 15, its rear was not configured with a cold high-pressure system of the same intensity as on March 15, which explains the difference in near-surface wind speed and direction in the dust source area (Fig. S11), thus regulating the dust transport path. The above analysis indicates that the

strength and location of the Mongolian cyclone played a key role in regulating the dust transport during these two events. On March 28, the Mongolian cyclone drifted further eastwards and the northwesterly winds on the west side of the cyclone transported the dust plume rapidly to the NCP. Up until March 29, the cyclone center was moving out of the Chinese region as a whole and dragged the dust plume to Northeast Asia.

### 3.6. Record-breaking regional dust loading in March 2021 over the past 20 years

The magnitude of DOD associated with these two events in March 2021 was a historic one and almost exceeded the climatology from the last 20 years, as recorded in the combined MODIS DOD data retrieved from both Terra and Aqua since 2000. First, a comparison was made between the combined March mean DOD in 2021 (Fig. 13a) and that of the 2000–2021 climatology (Fig. 13b). It should be emphasized that the combined DOD values here incorporate information from two sensors at different observation times, which is beneficial for improving the spatial coverage and is also more representative

than the DOD values at a single observation time. The results show that the combined DOD values were highly consistent with both Terra-based DOD and Aqua-based DOD in terms of spatial distribution, magnitude, and year-to-year variation (Figs. S12–S15). Furthermore, the combined DOD time series obtained from the two sensor retrievals also utilizes the expanding duration of the target dataset. Therefore, the analyses reported below were carried out based on the combined DOD record from 2001 to 2021.

Clearly, the magnitude of DOD in March 2021 was stronger than in the 20-year climatology, and this remarkable enhancement was mainly located in the GD and its downstream regions, including Ningxia, Gansu, the NCP, and Liaoning. As shown in Fig. 13c, the DOD in March 2020 was more than 0.2–0.8 (depending on the region) higher than the climatology. Among them, the largest positive DOD anomalies were in the Gobi sands and NCP regions. In contrast, a moderate negative anomaly was observed in the core area of the TD, which implies that the two processes had a weak impact on this region. To quantify the magnitude of the impact of these two events in different regions, a regional analysis of MODIS daily and monthly DODs since 2000 was carried out for the entire NC region and its four sub-regions as defined in Fig. 13c: northwest China (NWC), the GD, NCP, and northeast China (NEC). Results from the regional analysis on the daily and monthly scales are shown in Fig. 14 and Fig. 15, respectively. In each region, the daily regional-averaged DOD in March 2021 is highlighted as red dots and lines, while the gray lines depict in detail the evolution of the daily regional-averaged DOD in March for each year during the period of 2000–2020. Clearly, the 3.15 event had the highest or close to highest daily DOD over the past two decades over the entire NC region (March 16, Fig. 14a), the GD (March 15, Fig. 14c), and the NCP (March 18, Fig. 14d). In NWC, the magnitude of DOD is generally determined by the variability of local meteorological factors (mainly surface wind speed) (Che et al., 2019; Pu and Jin, 2021). Despite this, the enhanced DOD during the 3.15 event was still relatively high in the last 20 years, which indirectly reflects the wide range of regional impacts of the atmospheric circulation extremes.

In contrast, although the 3.27 event was weaker than the 3.15 event overall, the former still witnessed the largest or close to largest DOD on the same day in history over the entire NC region (March 28), the GD (March 27), the NCP (March 28), and NEC (March 28). The monthly regional analysis shows that March 2021 was also strongest or second strongest dust month over the past two decades over different study regions, which is attributable to the occurrence of these two dust events with historical levels of intensity in March 2021 (Fig.15). Specifically, in the GD (the entire NC region and NEC), March 2021 had the highest (second highest) DOD in March over the past two decades. In other words, March 2021 was the strongest dust month in these regions in the past 20 years, except for 2010.

The daily DOD anomalies in March 2021 were more prominent when focusing only on the combined period of March 15–20 and 27–29 (abbreviated as SDS days, Fig. 16 and Fig. 17). As shown in Fig. 16, the daily composite of SDS days in March 2021 is more than 1.0 higher than the climatology. Moreover, this DOD enhancement covers almost the entire northern region of China except for the northwestern part of Xinjiang, the northeastern part of Inner Mongolia, and the eastern part of Liaoning–Jilin–Heilongjiang. As expected, the daily composite of SDS days in March 2021 has the highest DOD in the past 20 years over the entire NC region, the GD, and the NCP (Fig. 17). Especially in the GD (the entire NC

region and the NCP), the magnitude of daily composited DOD on SDS days is almost twice (1.5 times) as high as the second highest (i.e., March 2010) in history.

### 3.7. Anomalous meteorological drivers conducive to dust emissions

So, what drove the record-breaking dust intensity in March 2021 over the past 20 years? Earlier, atmospheric circulation (i.e., Mongolian cyclone) extremes were identified as the main external driver of these two extreme SDS events. To determine how anomalous was the Mongolian cyclone during these two SDSs, we examined the intensity of the Mongolian cyclone in March and during the combined period of March 15 and 27 (representing the starting day of these two SDSs) from 2000 to 2021 (Fig. S16). Following Zhu et al. (2008), we defined the cyclone intensity by the 850 hPa GH averaged over the GD (black box in Fig. 18: 36°–47°N, 96°–112°W). Over the GD, the GH in 2021 is higher than the climatology by up to 54.4 gpm (Fig. S16a). In terms of ranking, the 2021 cyclone intensity over the GD is the highest over the last 2 decades. Moreover, the monthly regional analysis shows that such two anomalous Mongolian cyclones resulted in more than 6.7 gpm higher GH than climatology in March 2020. This analysis suggests that the two Mongolian cyclones in March 2021 was highly anomalous in intensity, which creates favorable dynamical conditions for the record-breaking regional dust loading in March 2021 over the past 20 years. Next, the internal drivers are explored by focusing on the extent of local meteorological anomalies affecting the surface conditions in the dust source region. Previous studies have revealed that the variability of dust emission intensity in the dust source area of NC is mainly controlled by several local meteorological factors, such as temperature, PPT and VSW (Kim and Choi, 2015; Wu et al., 2018; Yao et al., 2021). These factors have previously been demonstrated to constrain the intensity of dust emissions and their variability on multiple timescales.

Figs. 18a–d present the local meteorological anomalies of two weeks before the 3.15 event with reference to the 2000–2020 climatology, including the temperature at 2 m, SD, PPT, and VSW. Since the two strong SDS processes in March 2021 originated from the GD, the annual time series of these meteorological factors averaged over the GD (black box in Fig. 18) were further calculated, as shown in Figs. 18e–h. The results of the 3.27 event are shown in Fig. S17. The ERA5 meteorological analysis indicates that, from the beginning of March 2021, the near-surface temperature in western Inner Mongolia and Mongolia was more than 4.0 °C warmer than the climatology (Fig. 18a), leading to early melting of snow covering the ground (Fig. 18b) and thus further caused the ground to become bare and loose. The high surface temperature also indicated a strong ground surface evaporation, which accelerated land drying. Meanwhile, the dust source areas tended to become drier owing to decreased VSW associated with negative precipitation anomalies (Figs. 18c and d). On the one hand, such a high temperature anomaly made it easier to form a warm low pressure on the ground, the pressure of the Mongolian cyclone was likely to be lower, and the pressure gradient between the cold and high pressure was likely to be larger, which would have been conducive to aggravating gales. On the other hand, such a dry environment would have made the surface of the sand source looser, which would have favored an enhancement of dust emissions driven by the strong winds. Although the magnitude of these local meteorological anomalies was moderate before the 3.27 event (Fig. S17), their

overall pattern did not change significantly, which to some extent created favorable conditions for the 3.27 event to occur. The climatic anomalies of these local meteorological factors in the dust source area may be closely related to the anomalies of sea ice shift in the Barents and Kara Sea, and sea surface temperatures in the east Pacific and northwest Atlantic (Yin et
al., 2021).

The time series of regional-averaged local meteorological drivers suggest that, while an exceptionally strong Mongolian cyclone triggered dust emissions from the GD, intensified temperatures (4.6°C warmer than the climatology and the strongest in the past 22 years for March 1–14) led to the melting of snow (1.85 mm lower than the climatology and the lowest in the past 22 years), accompanied by decreased PPT (0.007 mm lower than the climatology and the second lowest in
the past 22 years) and VSW (0.011 $m^3\,m^{-3}$ lower than the climatology and the third lowest in the past 22 years) systematically contributed to the further enhancement of dust emissions during the 3.15 event. For the 3.27 event, positive anomalies significantly above or below the climatology were observed for all four meteorological factors (Fig. S17).

## 4. Conclusions and implications

In March 2021, two unexpected mega SDS events (referred to here as the "3.15" and "3.27" events), separated by only a
625 week, invaded most of NC, significantly worsening the air quality, threatening people's health and disrupting economic and social activities. In this study, the sun photometer observations in Beijing and a comprehensive set of multiple satellite and ground-based observations combined with atmospheric reanalysis data were used to characterizes the optical, microphysical, and radiative properties of aerosols and their meteorological drivers during these two SDS events. Meanwhile, the historical ranking of the dust loading in NC during dust events was evaluated by using a long-term (2000–2021) DOD dataset retrieved
from MODIS measurements.

During both events, the invasion of dust plumes greatly worsened the horizontal visibility over large areas of NC, with extreme low visibility levels ranging from tens to hundreds of meters recorded at several sites. Despite the shorter duration of the 3.27 event relative to the 3.15 event, sun photometer and satellite observations in Beijing recorded a larger peak AOD (~2.5) in the former than in the latter (~2.0), which was mainly attributed to the short-term intrusion of coarse-mode dust
particles with larger effective radii (~1.9 μm) and volume concentrations (~2.0 $μm^3\,μm^{-2}$) during the 3.27 event. Such strong dust plumes have a non-negligible impact on the short-term radiation balance. The shortwave DARF induced by dust was estimated to be −92.1 and −111.4 W $m^{-2}$ at the top of the atmosphere, −184.7 and −296.2 W $m^{-2}$ at the surface, and +92.6 and +184.8 W $m^{-2}$ in the atmosphere in Beijing during the 3.15 and 3.27 event, respectively. Analysis of MERRA-2 dust emissions and Himawari-8 dust RGB composite images suggested that the two SDS processes originated in the GD across
southern Mongolia and NC. Specifically, mineral dust aerosols from the GD in Mongolia were first uplifted and subsequently mixed with those emitted from the GD in NC, and were continually transported to the downstream region. During the dust transport episode of the 3.15 event, the dust plume with an DEC greater than 0.05 $km^{-1}$ was lifted to an altitude of 4–8 km, and its range of impact extended from the dust source to the eastern coast of China. In contrast, the lifting altitude of the dust

plume during the 3.27 event was lower than that during 3.15 event, with the top of the dust plume (DEC > 0.05 km$^{-1}$) at about 5 km. The difference in lifting altitude of the dust plume during these two dust transport processes was also confirmed by ground-based lidar observations in Inner Mongolia. For these two mega SDS events, the regional-averaged DOD in the dust source (here, the GD) and its downstream (i.e., the NCP) broke the MODIS record in the past 20 years for the same period (i.e., the combined period of March 15–20 and 27–29) in history, with the daily mean DOD exceeding 2.0 over the GD and NCP. On a monthly average scale, the strong impacts attributed to these two processes directly led to March 2021 being registered as the strongest DOD month in the past decade across the entire NC region, even in the past two decades, second only to March 2010 (Bian et al., 2011; Tan et al., 2017).

Analysis of ERA5 meteorological data suggested that these two mega SDS events were associated with both atmospheric circulation extremes and local meteorological anomalies that favored enhanced dust emissions in the GD. Firstly, both SDS events were caused by strong surface wind speeds triggered by an exceptionally strong Mongolian cyclone generated at nearly the same location (along the central and eastern plateau of Inner Mongolia) in conjunction with the surface-level cold high-pressure system at the rear. Secondly, although anomalies in surface wind speed provided the dynamical conditions for dust emissions, the early melting of spring snow caused by near-surface temperature anomalies over dust source regions, together with the negative soil moisture anomalies induced by decreased precipitation, formed drier and barer soil surfaces, which systematically provided the material conditions for the SDS events to occur. Although the atmospheric circulation anomalies in both events were similar to the typical circulation patterns that facilitate the occurrence of spring SDS events in NC (Zhu et al., 2008), the degree of surface dryness/bareness and wind anomalies were astounding, emphasizing the substantial contribution of the joint effects of the surface condition and atmospheric circulation anomalies to the occurrence of both extreme SDS events.

Against the backdrop of the continued absence of strong SDS events in NC in almost a decade (An et al., 2018; Wang et al., 2018; Liu et al., 2020; Yao et al., 2021), this unexpected resurgence of two mega SDS events has raised potential concern as to whether such extreme SDS events will occur frequently in the future or whether a fresh active cycle of dust will begin. Currently, there is no consensus on whether future dust aerosol emissions in NC will increase or decrease. Some studies suggest that dust emissions and the occurrence frequency of SDS events in NC may continually decrease in the future (Tegen et al., 2004; Liu et al., 2020; Pu and Ginoux, 2016), and they attribute this to enhanced Arctic amplification under the future climate (Liu et al., 2020) accompanied by reduced temperature gradients at mid and high latitudes leading to reduced westerly winds, increased precipitation and enhanced leaf area index (Pu and Ginoux, 2016), which is not conducive to dust emissions. However, other studies have also found that, under a scenario of continued global warming, land degradation and desertification in arid areas of East Asia will be aggravated, and the inner region of East Asia (covering the GD) is likely to become drier and hotter (Huang et al., 2016; Zhang et al., 2020; Zong et al., 2021), which will provide favorable surface conditions for increases in future dust emissions. Given the importance of Asian dust in regional climate, ecosystems, environment, air quality, and public health, further exploration is still needed in the future regarding how dust aerosols may evolve in NC. The present study highlights that improving the projection of large-scale circulation anomalies (especially

Mongolian cyclones) and surface conditions will be the key determinant in terms of confidence in climate models to predict whether dust aerosols in NC will increase or decrease in the future.

**Data availability.** Aerosol optical, microphysical and radiative properties data retrieved by the sun photometer observations in Beijing used in this study can be requested by contacting the corresponding author. Other data or products used in this study were obtained from various publicly available sources: the MODIS aerosol optical property data (MOD04_L2 and MYD04_L2) and VIIRS AOD products (AERDB_L2 and AERDT_L2) were obtained from Earthdata Search (https://search.earthdata.nasa.gov/search), a web application developed by NASA's Earth Observing System Data and Information System (EOSDIS). The CALIOP aerosol extinction profile product (CAL_LID_L2_05kmAPro-Standard-V4-21) was obtained from the NASA Langley Atmospheric Science Data Center (ASDC) by CALIPSO's search and subsetting web application (https://subset.larc.nasa.gov/calipso/login.php). The Himawari-8 product is available at ftp.ptree.jaxa.jp. The MERRA-2 aerosol reanalysis data are available via the Goddard Earth Sciences Data and Information Services Center (GES DISC) (https://earthdata.nasa.gov/eosdis/daacs/gesdisc). The ERA5 reanalysis data from ECMWF can be accessed at the Copernicus Climate Data Store (CDS) (https://cds.climate.copernicus.eu/cdsapp#!/home). The hourly horizontal visibility data were obtained from the National Meteorological Information Center (http://data.cma.cn/site/index.html) of the CMA. The ground-based lidar data were provided courtesy of AD-Net (https://www-lidar.nies.go.jp/AD-Net).

**Author contributions.** HC and KG designed the study. KG performed the data analysis with contributions from all co-authors; KG and WY prepared and drafted the paper with help from HC and LA; LL, YZ, HZ, LZ, JZ, YW, and XZ provided constructive suggestions on this study.

**Competing Interests.** The authors declare that they have no conflict of interest.

**Acknowledgements.** This research has been supported by the National Science Fund for Distinguished Young Scholars (grant no. 41825011), National Key R&D Program Pilot Projects of China (grant no. 2016YFA0601901), the National Natural Science Foundation of China project (grant nos. 42175153, 42030608 and 41905121), and the Basic Research Fund of CAMS (No. 2021Y001).

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

**Table 1. Daily arithmetic mean of aerosol optical/microphysics parameters at Beijing-CAMS site during the two SDS events.**

| Day | 3.15 event | | | 3.27 event | |
|---|---|---|---|---|---|
| | Mar. 16 | Mar. 17 | Mar. 20 | Mar. 28 | Mar. 29 |
| $N_{inst}$ [a] | 3 | 3 | 4 | 1 | 1 |
| $SSA_{440nm}$ [b] | $0.984 \pm 0.002$ | $0.915 \pm 0.025$ | $0.950 \pm 0.042$ | 0.900 | 0.986 |
| $AAOD_{440nm}$ [b] | $0.004 \pm 0.001$ | $0.164 \pm 0.047$ | $0.016 \pm 0.014$ | 0.249 | 0.002 |
| $AAE_{440-870nm}$ [c] | $0.849 \pm 0.217$ | $2.973 \pm 0.269$ | $1.345 \pm 1.078$ | 2.799 | 0.843 |
| $R_{eff_t}$ (μm) [b] | $0.738 \pm 0.044$ | $0.940 \pm 0.061$ | $1.305 \pm 0.234$ | 1.019 | 0.878 |
| $R_{eff_f}$ (μm) [b] | $0.126 \pm 0.010$ | $0.157 \pm 0.018$ | $0.113 \pm 0.007$ | 0.080 | 0.146 |
| $R_{eff_c}$ (μm) [b] | $1.727 \pm 0.057$ | $1.661 \pm 0.058$ | $2.360 \pm 0.200$ | 1.850 | 1.722 |
| $Volume_t$ ($\mu m^3 \mu m^{-2}$) [b] | $0.136 \pm 0.008$ | $1.110 \pm 0.066$ | $0.270 \pm 0.053$ | 2.084 | 0.081 |
| $Volume_f$ ($\mu m^3 \mu m^{-2}$) [b] | $0.014 \pm 0.001$ | $0.088 \pm 0.010$ | $0.011 \pm 0.002$ | 0.076 | 0.007 |
| $Volume_c$ ($\mu m^3 \mu m^{-2}$) [b] | $0.121 \pm 0.007$ | $1.022 \pm 0.056$ | $0.259 \pm 0.054$ | 2.007 | 0.074 |

[a] Number of instantaneous observations. [b] Optical parameters at a wavelength of 440 nm. [c] Absorption angström exponent between 440 and 870 nm.

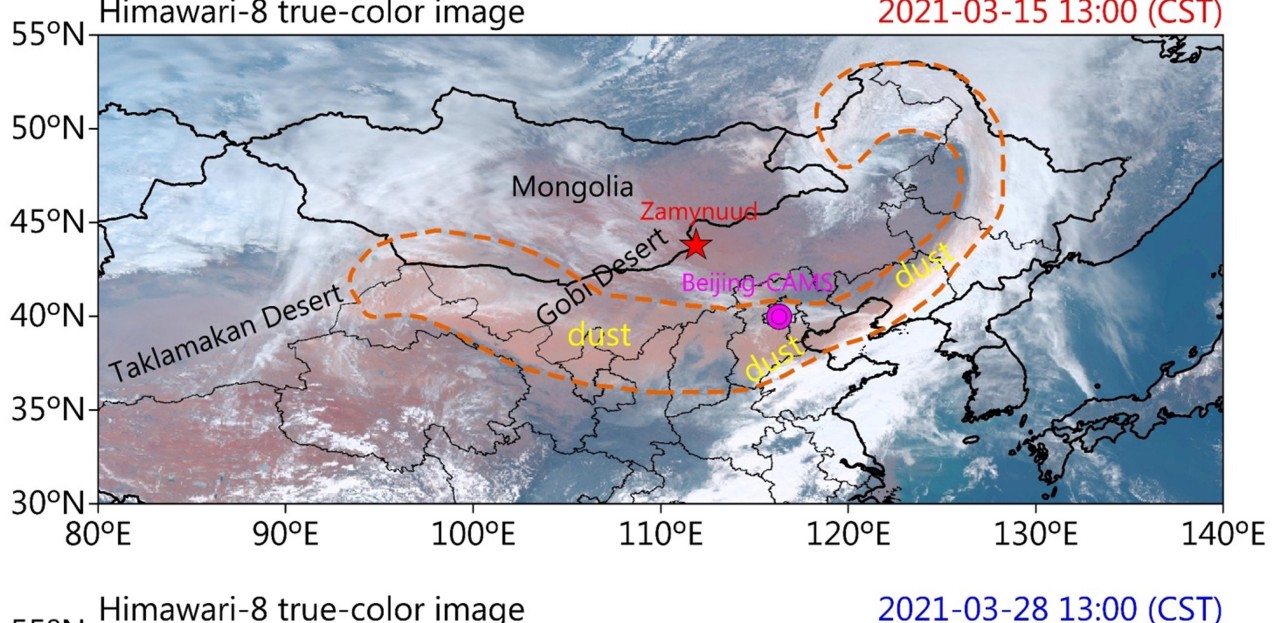

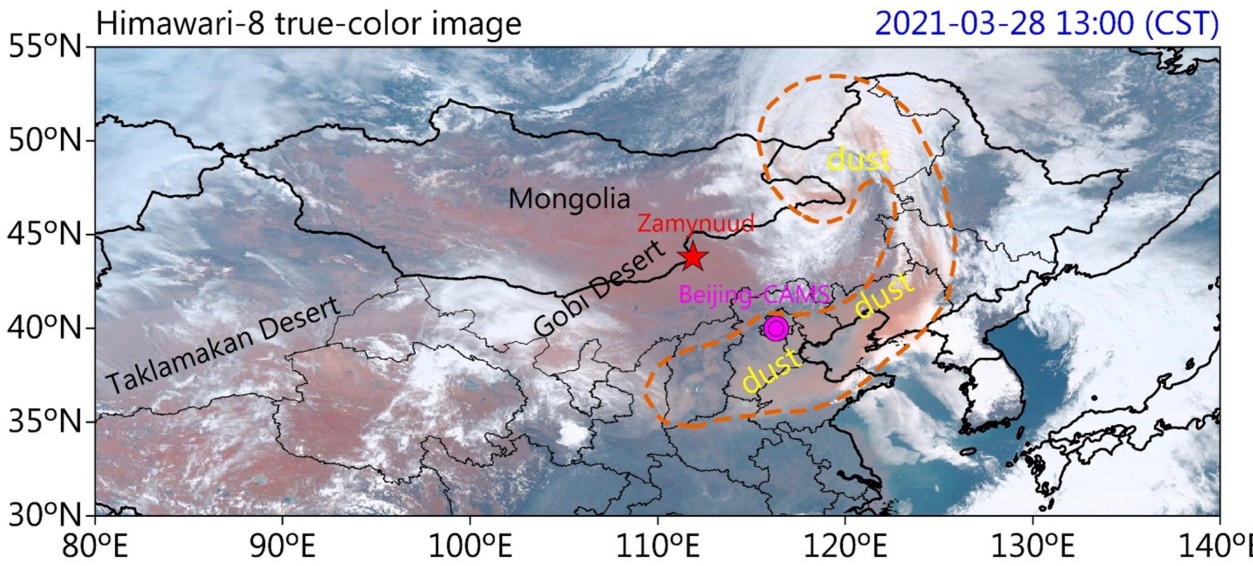

**Figure 1: True-color image of dust plumes above the Earth's surface captured by the Himawari-8 at 13:00 CST (China standard time) of March 15 (top) and March 28 (bottom), 2021. The location of the AD-Net Lidar site named "Zamynuud" (43.72°N, 111.90°E; 962 m) and the sun photometer site named "Beijing-CAMS" (39.93°N, 116.32°E; 106 m) are marked on the map with a red star and a magenta circle, respectively. Orange dashed lines outline the area covered by the dust plume.**

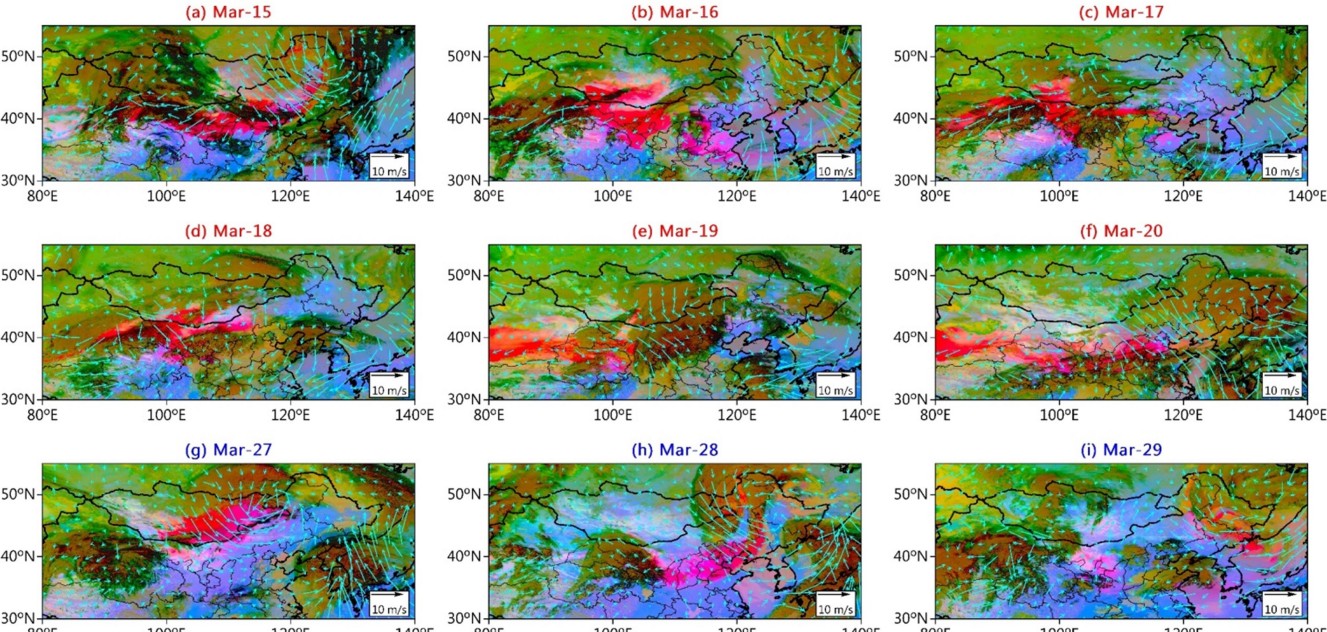

**Figure 2: Evolution of dust plumes (magenta) as revealed by Himawari-8 dust RGB composite images at 13:00 CST during (a–f) the 3.15 SDS event (March 15–20, 2021) and (g–i) the 3.27 SDS event (March 27–29, 2021), respectively. Overlaid on the RGB imagery is the ERA5 daily mean wind vectors at 10m.**

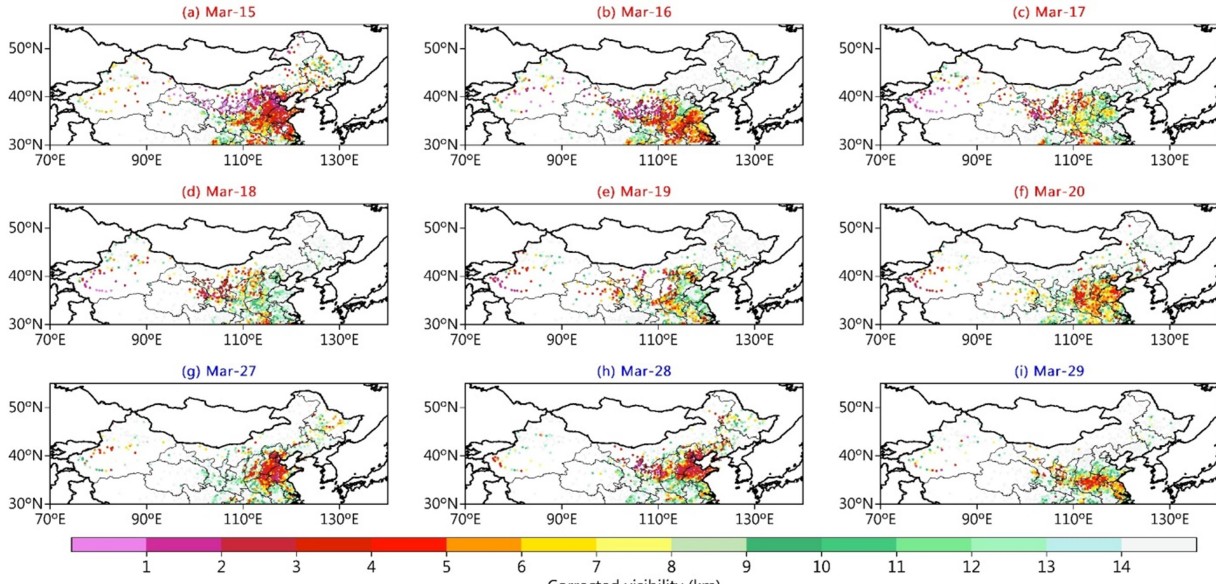

**Figure 3: Evolution of observed daily mean (presented as averages close to the MODIS and VIIRS observation time range, i.e., approximately 10:00 to 14:00 CST) corrected visibility during (a–f) the 3.15 SDS event (March 15–20, 2021) and (g–i) the 3.27 SDS event (March 27–29, 2021), respectively.**

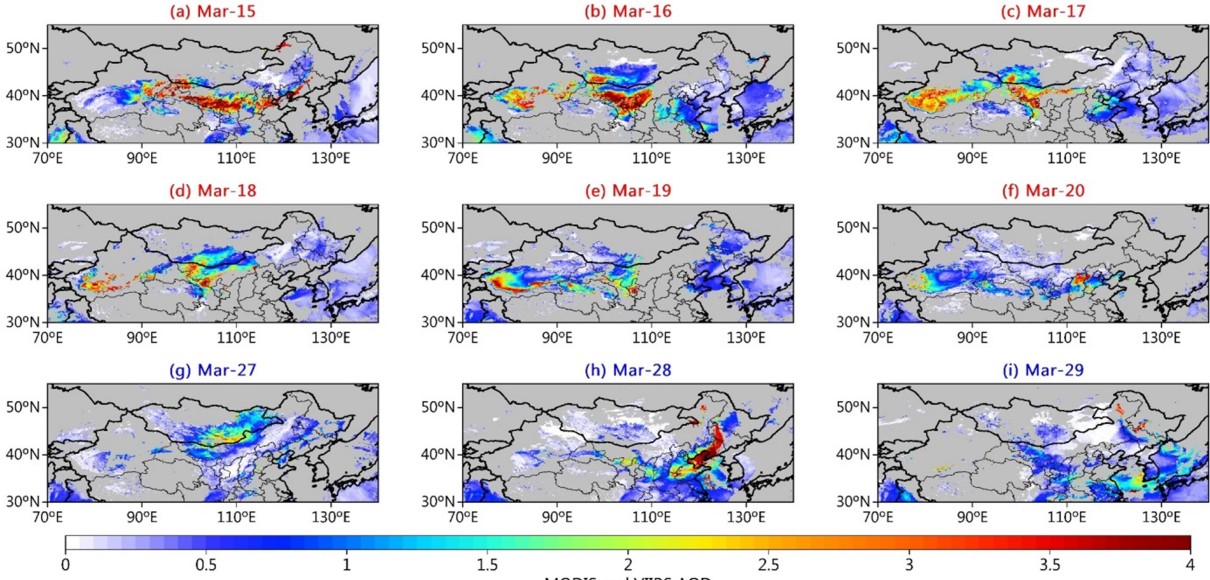

Figure 4: Evolution of MODIS and VIIRS combined daily mean AOD during (a–f) the 3.15 event and (g–i) the 3.27 event.

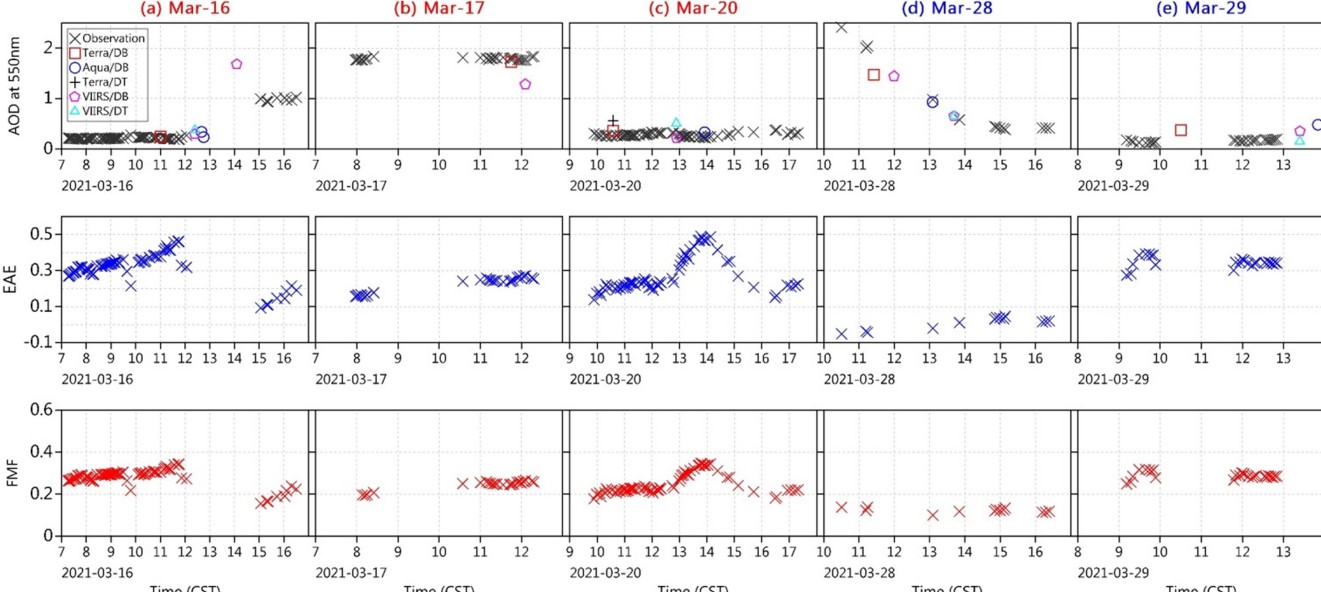

**Figure 5: Daily variation of AOD at 550 nm (top row), EAE between 440 and 870 nm (middle row), and FMF (bottom row) at Beijing-CAMS site during (a–c) the 3.15 event and (d-e) the 3.27 event. The instantaneous AOD values from the MODIS and VIIRS sensors, which were derived using the Dark Target (DT) and Deep Blue (DB) algorithms, respectively, are given in the top panel.**

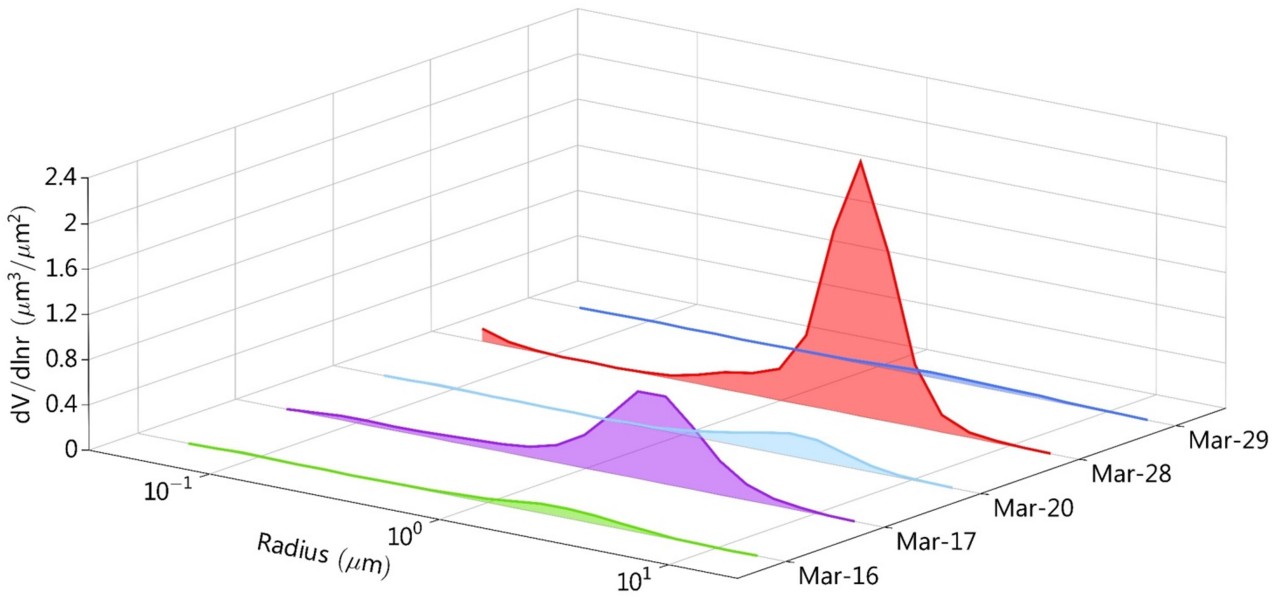

**Figure 6: Daily variation of the aerosol volume-size distributions at Beijing-CAMS site during the two SDS events.**

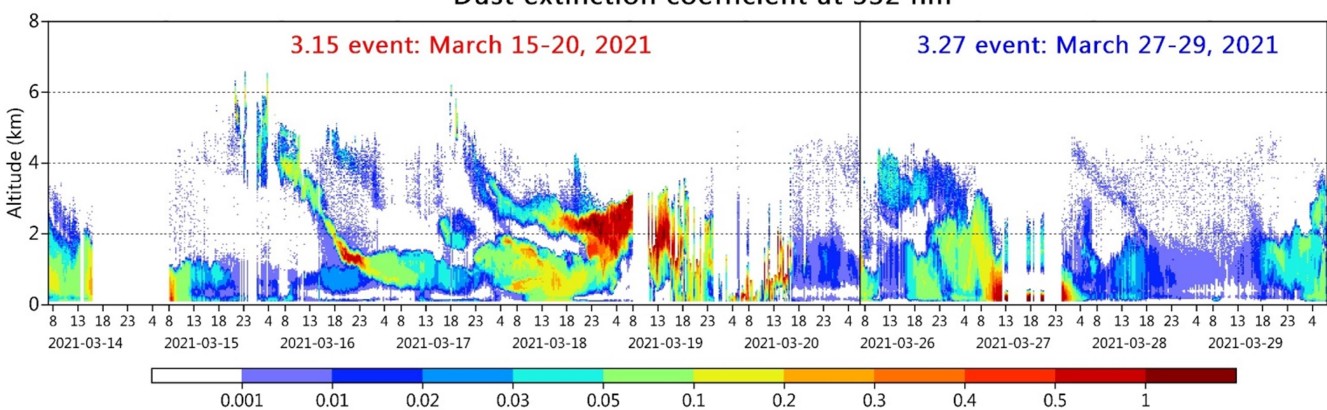

**Figure 7:** **Time–height evolution of dust extinction coefficient (km$^{-1}$) at 532 nm retrieved by ground-based Lidar (location of the site shown in Fig. 1) during the 3.15 and 3.27 events.**

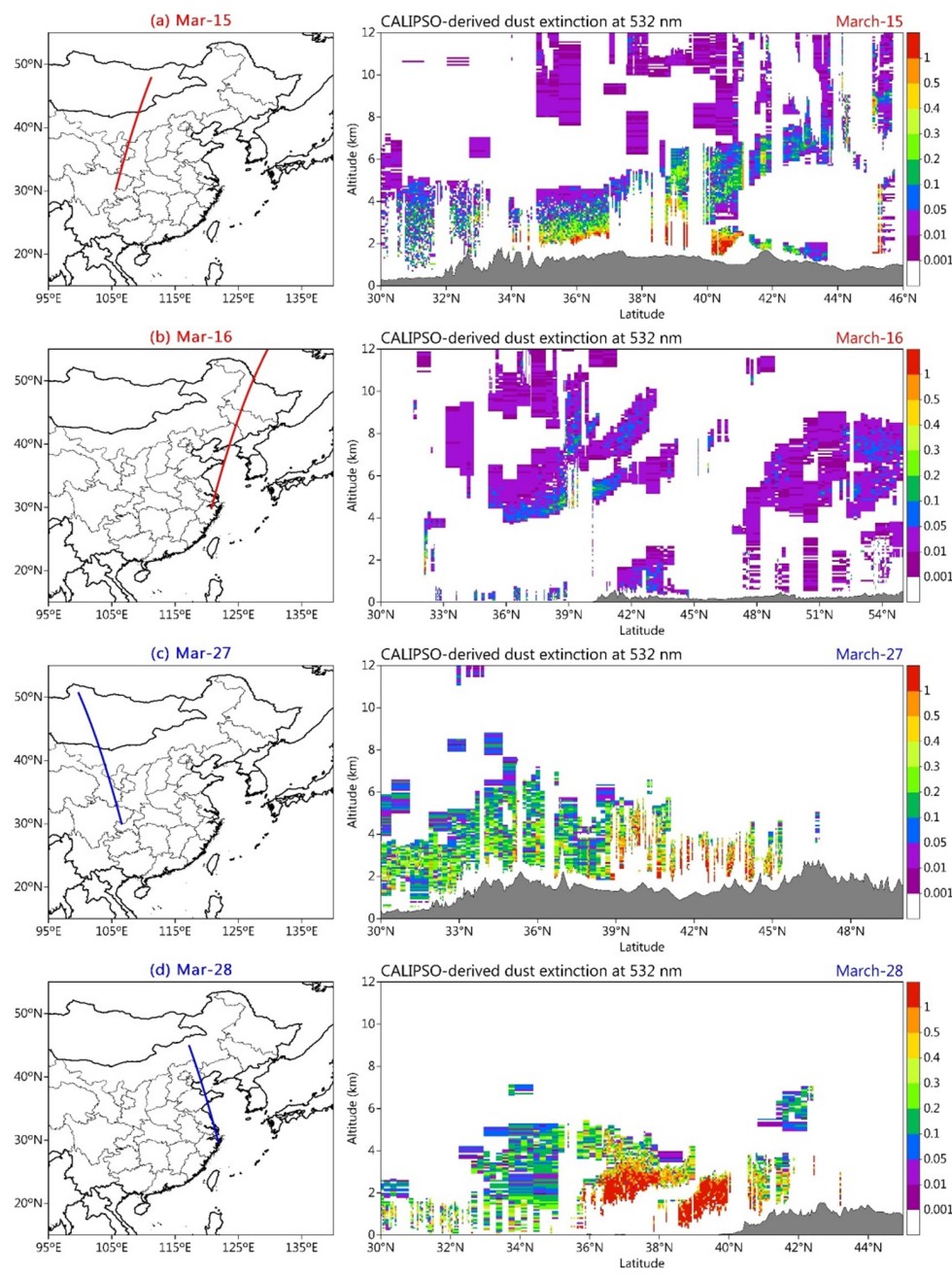

**Figure 8: CALIOP/CALIPSO snapshots of dust plumes for (a) March 15, (b) March 16, (c) March 27, and (d) March 28, 2021. The first column presents the CALIPSO tracks (red or blue lines) and the second column shows the 532 nm dust extinction coefficients (km$^{-1}$). Surface elevation is indicated by the gray filled line.**

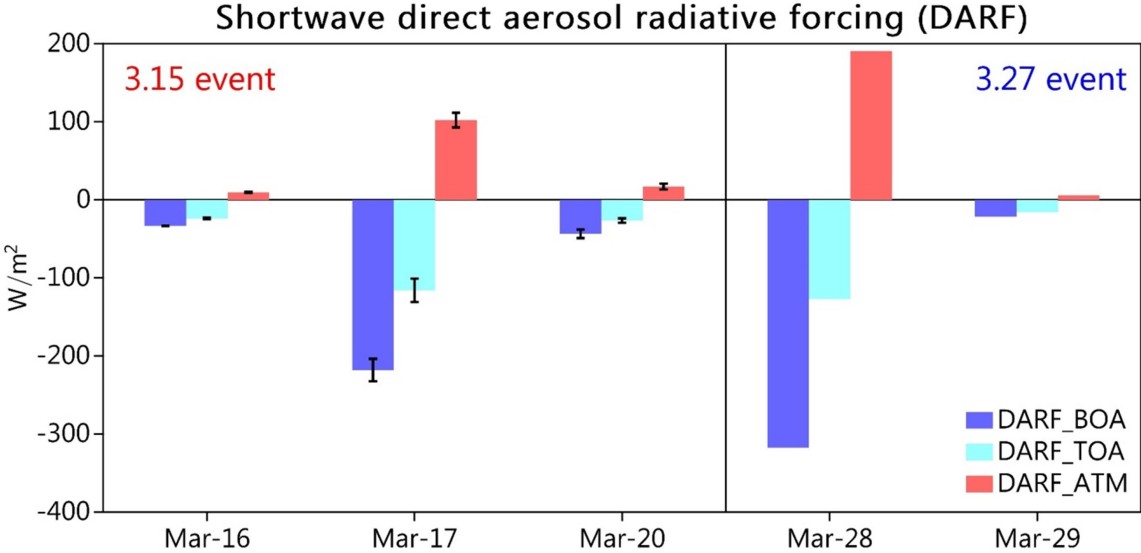

**Figure 9: Daily variation of the shortwave direct aerosol radiative forcing (DARF) at Beijing-CAMS site during the two SDS events.**

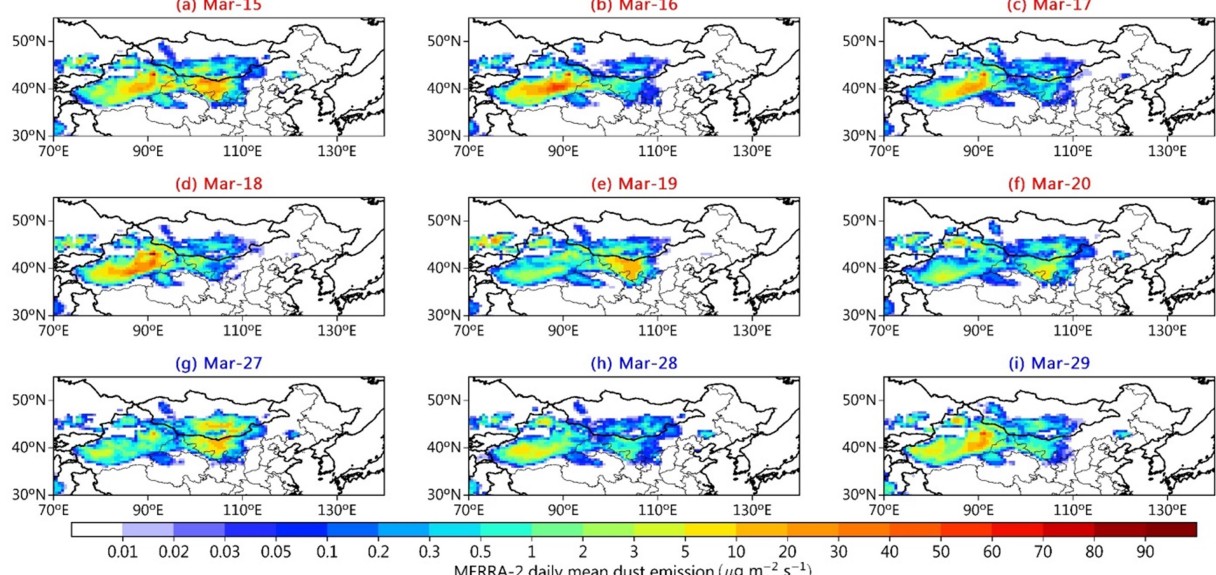

**Figure 10: Evolution of MERRA-2 daily mean dust emissions for all size bins during (a–f) the 3.15 event and (g–i) the 3.27 event.**

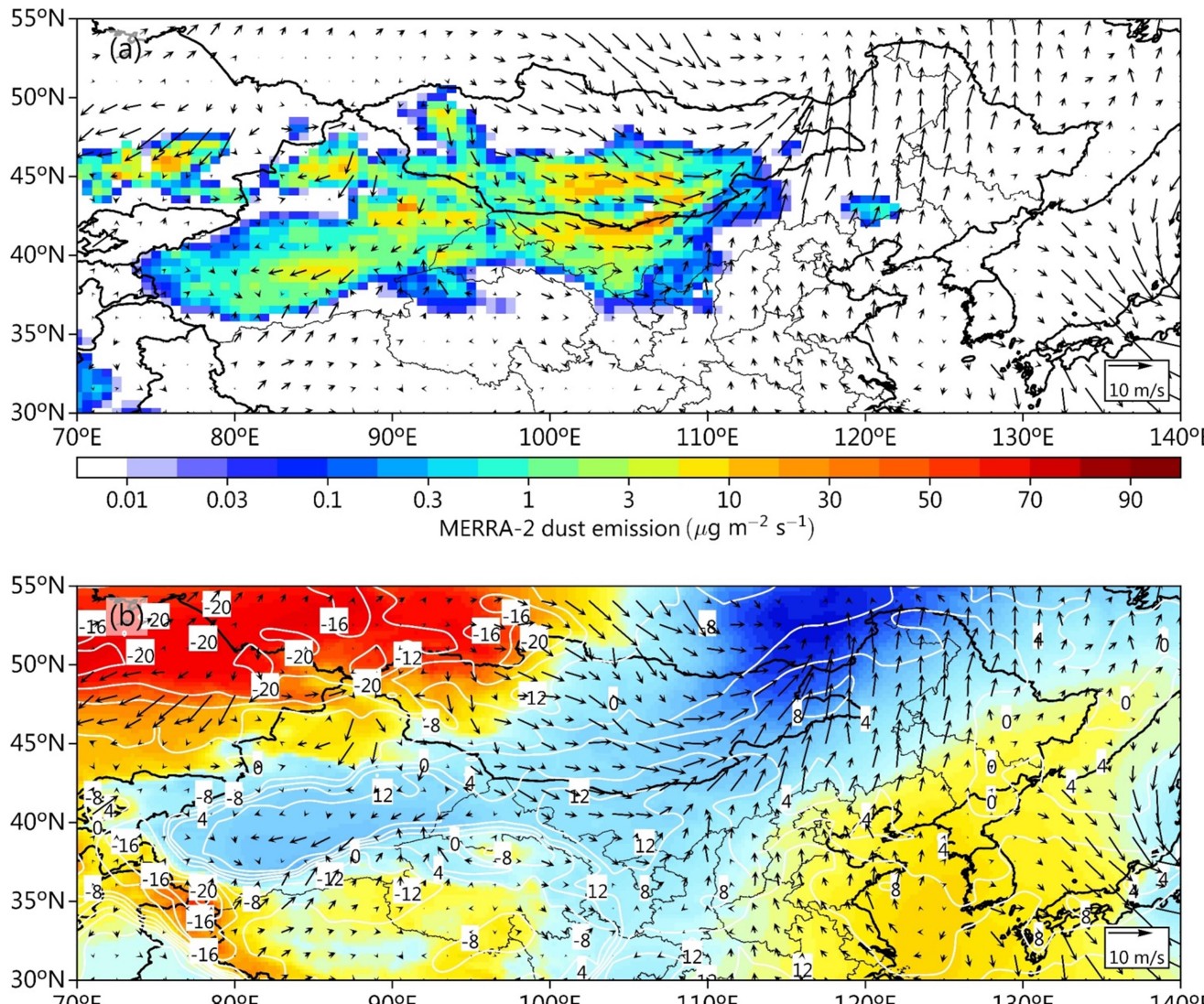

**Figure 11: (a) MERRA-2 daily mean dust emissions for all size bins on March 14, 2021. (b) Daily mean sea level pressure (SLP, shading) and temperature at 2m (contour; °C) on March 14, 2021. Overlaid on (a, b) are the ERA5 wind vectors at 10 m.**

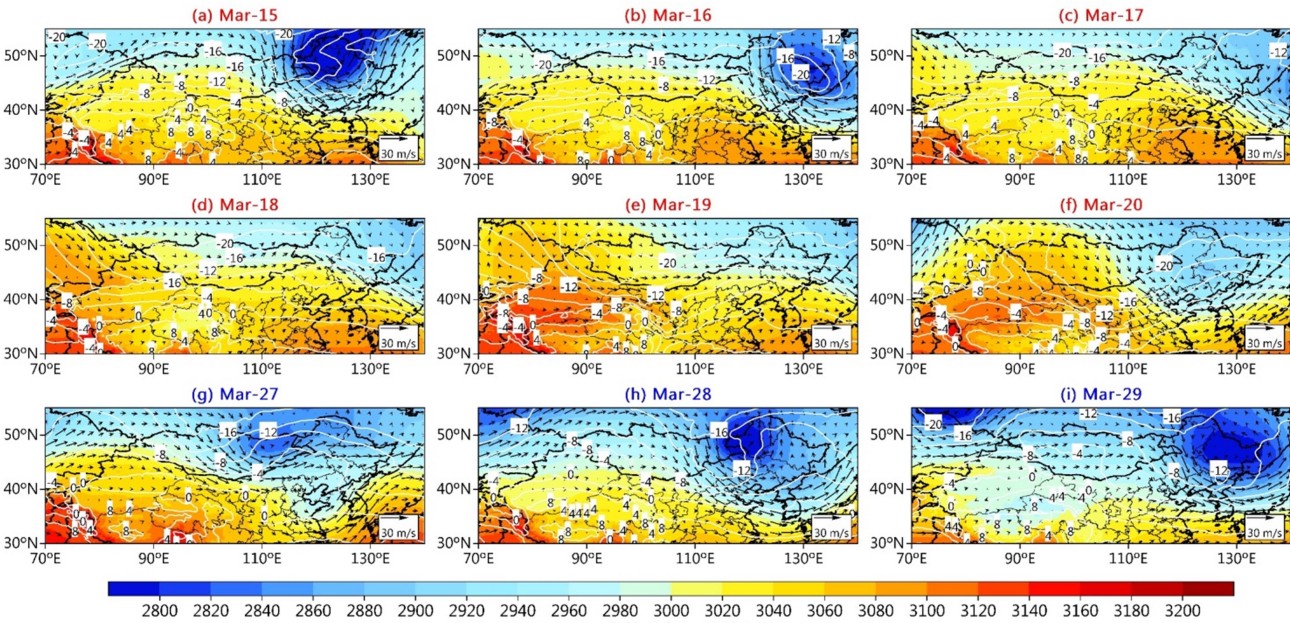

**Figure 12: Pattern evolutions of ERA5 geopotential height (shading; gpm), temperature (contours; °C), and wind vectors (black arrows; m s$^{-1}$) at 700 hPa on (a) March 15, (b) 16, (c) 17, (d) 18, (e) 19, (f) 20, (g) 27, (h) 28, and (i) 29, 2021.**

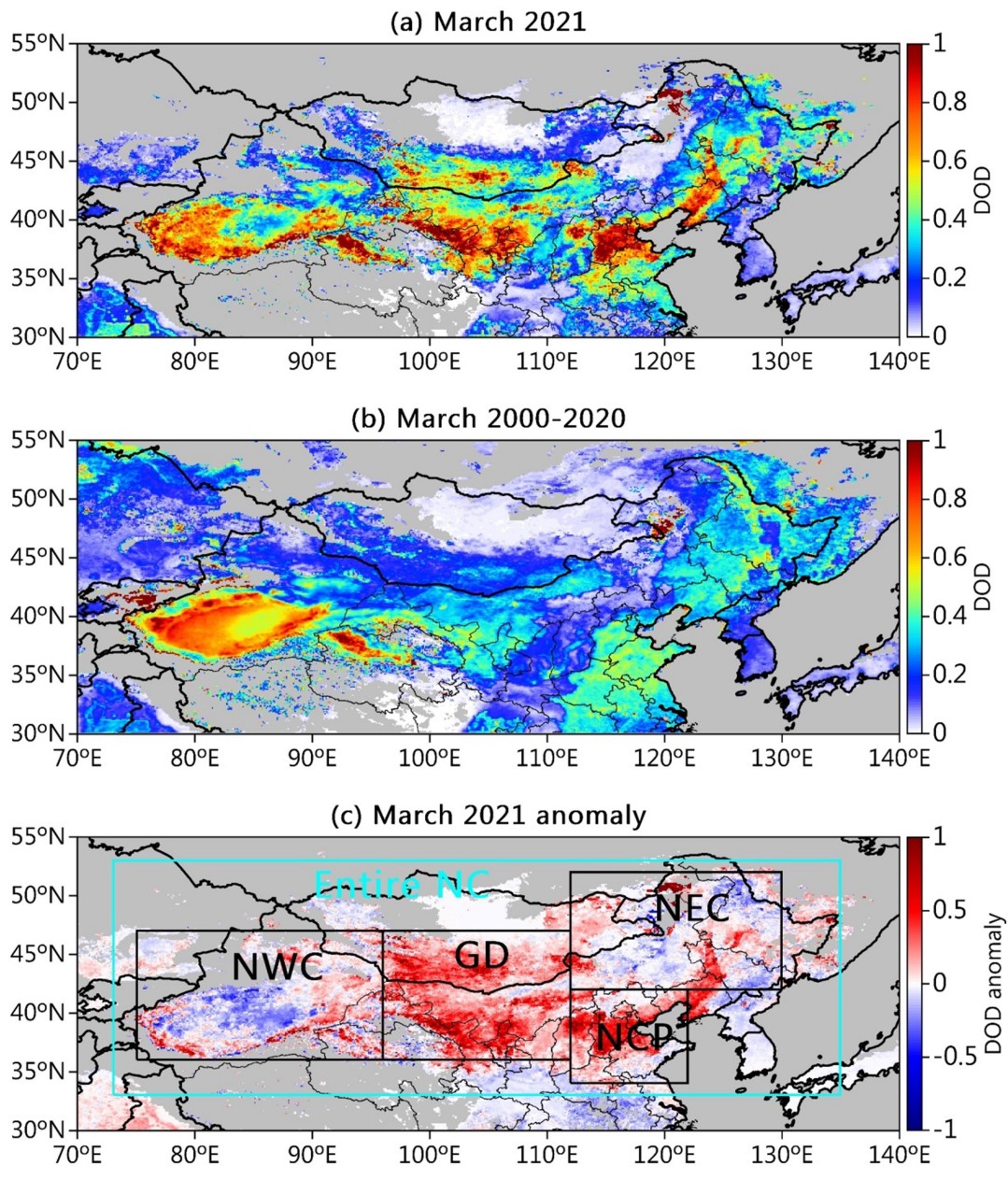

**Figure 13: MODIS-retrieved DOD: (a) March 2021, (b) March climatology (2000–2020), and (c) March 2021 anomaly. Cyan and black boxes indicate the averaging areas for the DOD time series, including (a) the entire northern China region (entire NC; 33°–53°N, 73°–135°W), (b) northwest China (NWC; 36°–47°N, 75°–96°W), (c) the Gobi Desert (GD; 36°–47°N, 96°–112°W), (d) the North China Plain (NCP; 34°–42°N, 112°–122°W), and (e) northeast China (NEC; 42°–52°N, 112°–130°W).**

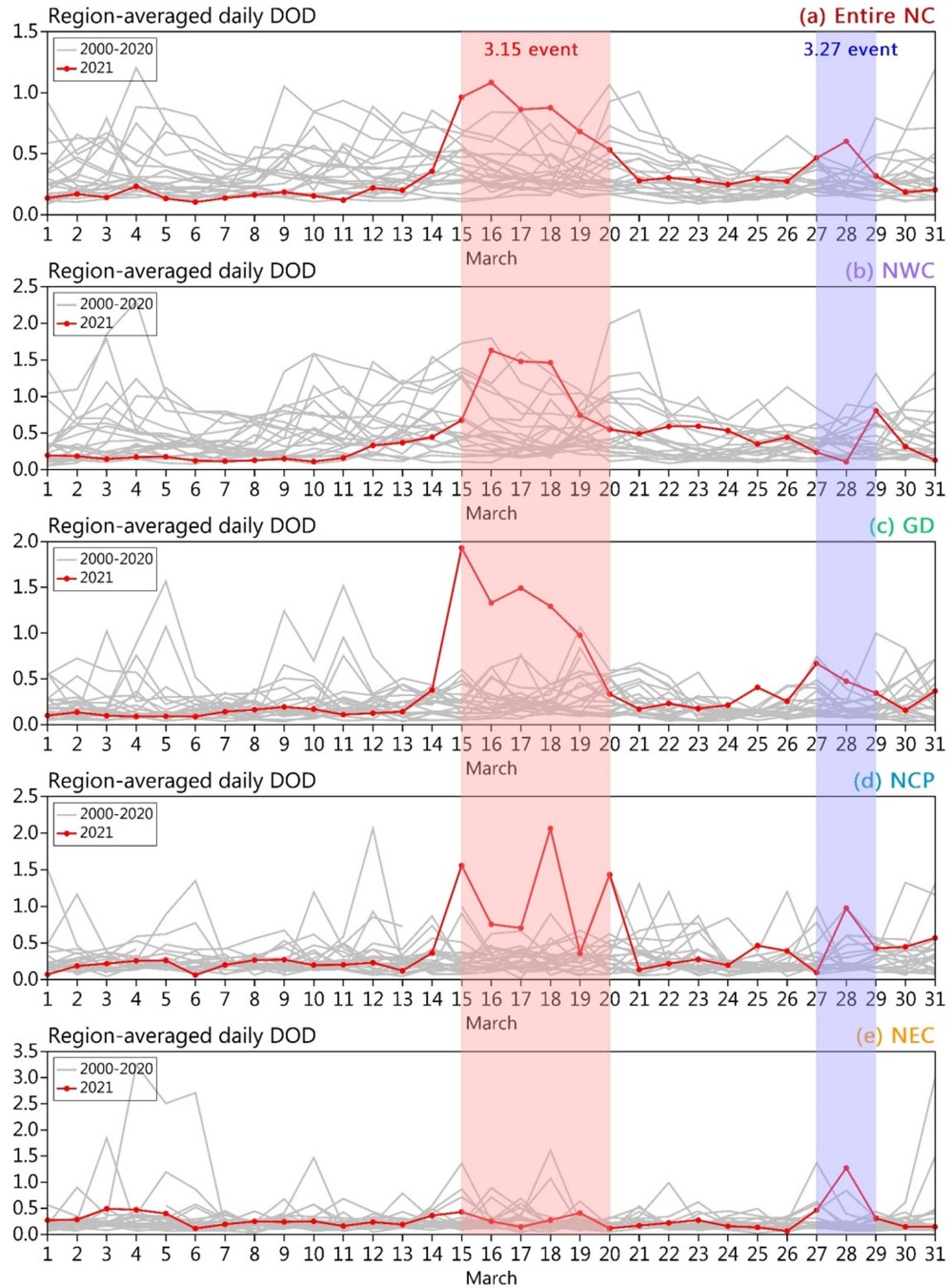

**Figure 14: MODIS-retrieved daily mean DOD for March 2021 (thick lines with red dots) in comparison to the 2000–2020 climatology (the year-to-year fluctuation range of daily DOD is represented by the thick gray line) in five regions, as defined in Fig. 13c. The days covered by the 3.15 and 3.27 events are marked in red and blue shading, respectively.**

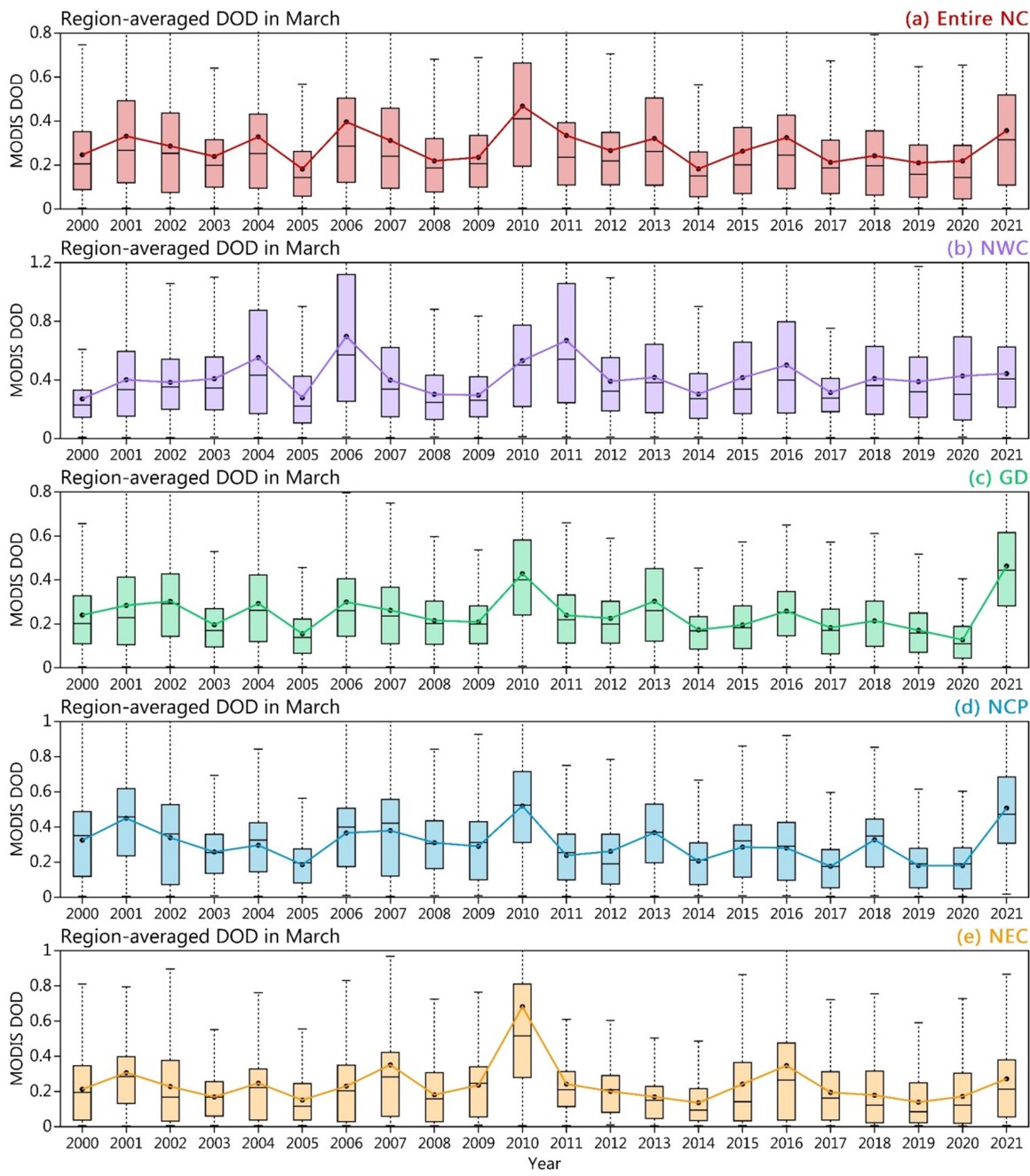

**Figure 15:** Time-series boxplots of MODIS-retrieved regional-averaged DOD over (a) the entire NC region, (b) NWC, (c) the GD, (d) the NCP, and (e) NEC in March from 2000 to 2021.

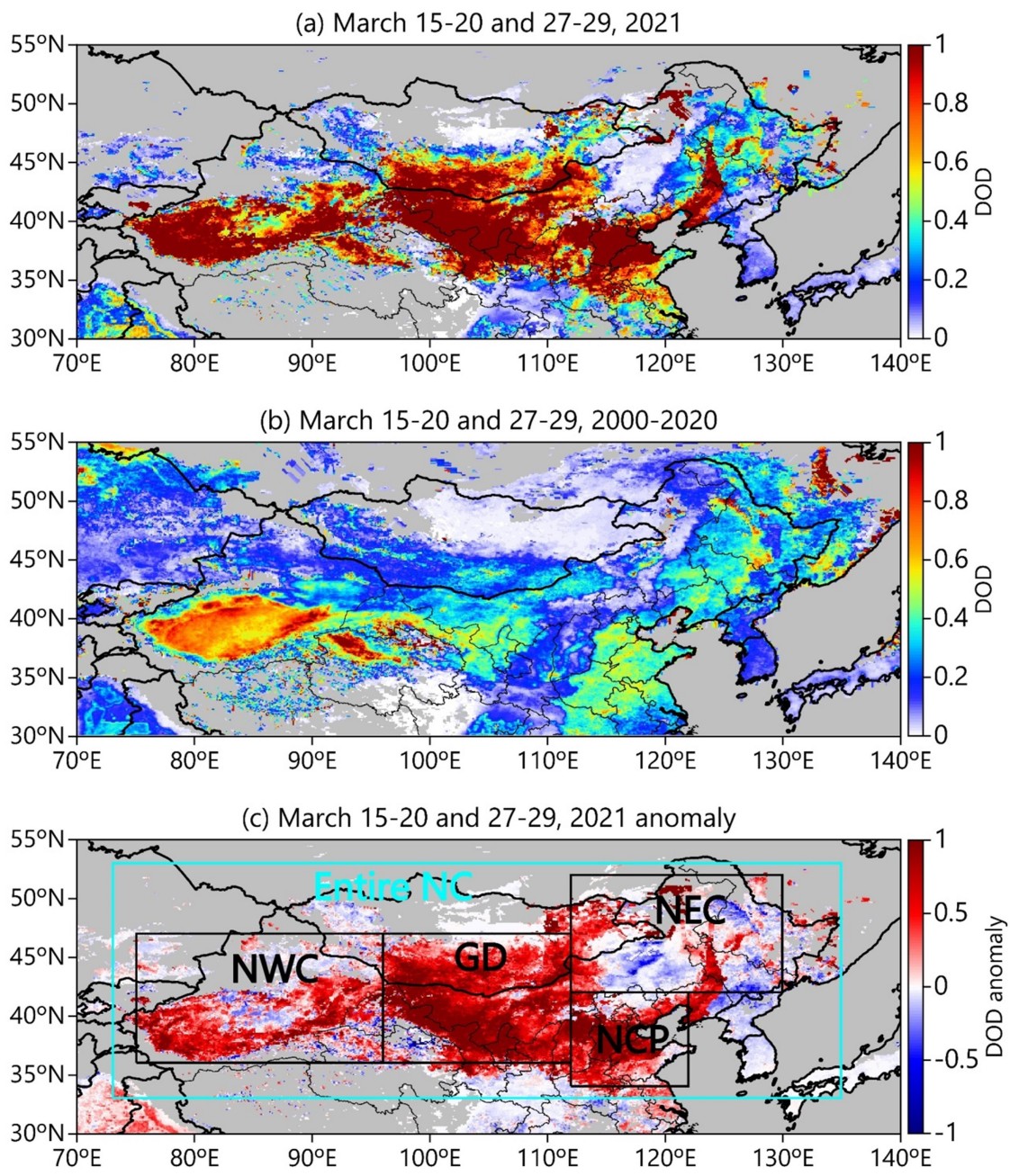

**Figure 16: As in Fig. 13 but for the combined MODIS-retrieved DOD from Terra and Aqua during the combined period of March 15–20 and 27–29.**

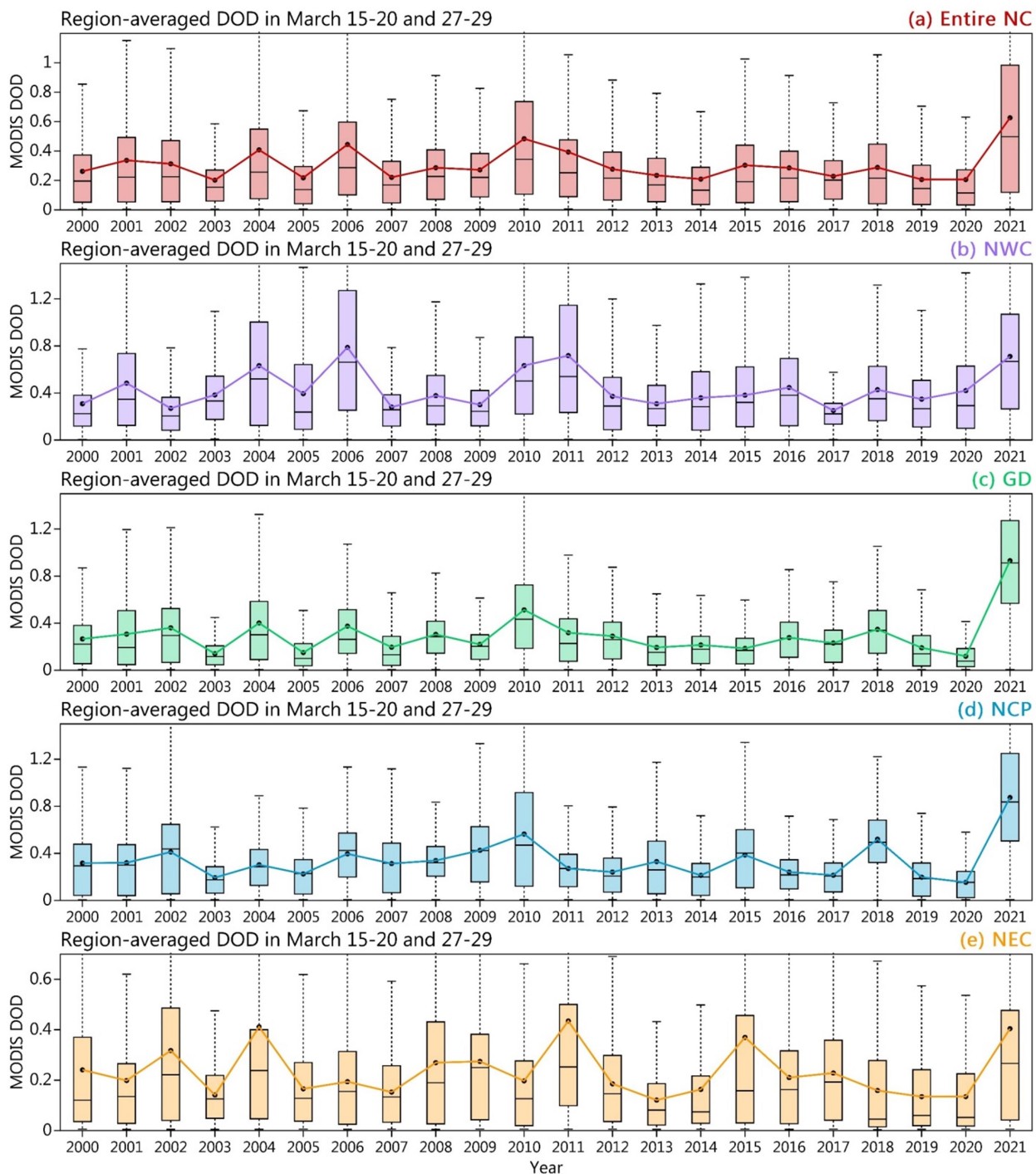

**Figure 17: As in Fig. 15 but for the combined MODIS-retrieved DOD from Terra and Aqua during the combined period of March 15–20 and 27–29.**

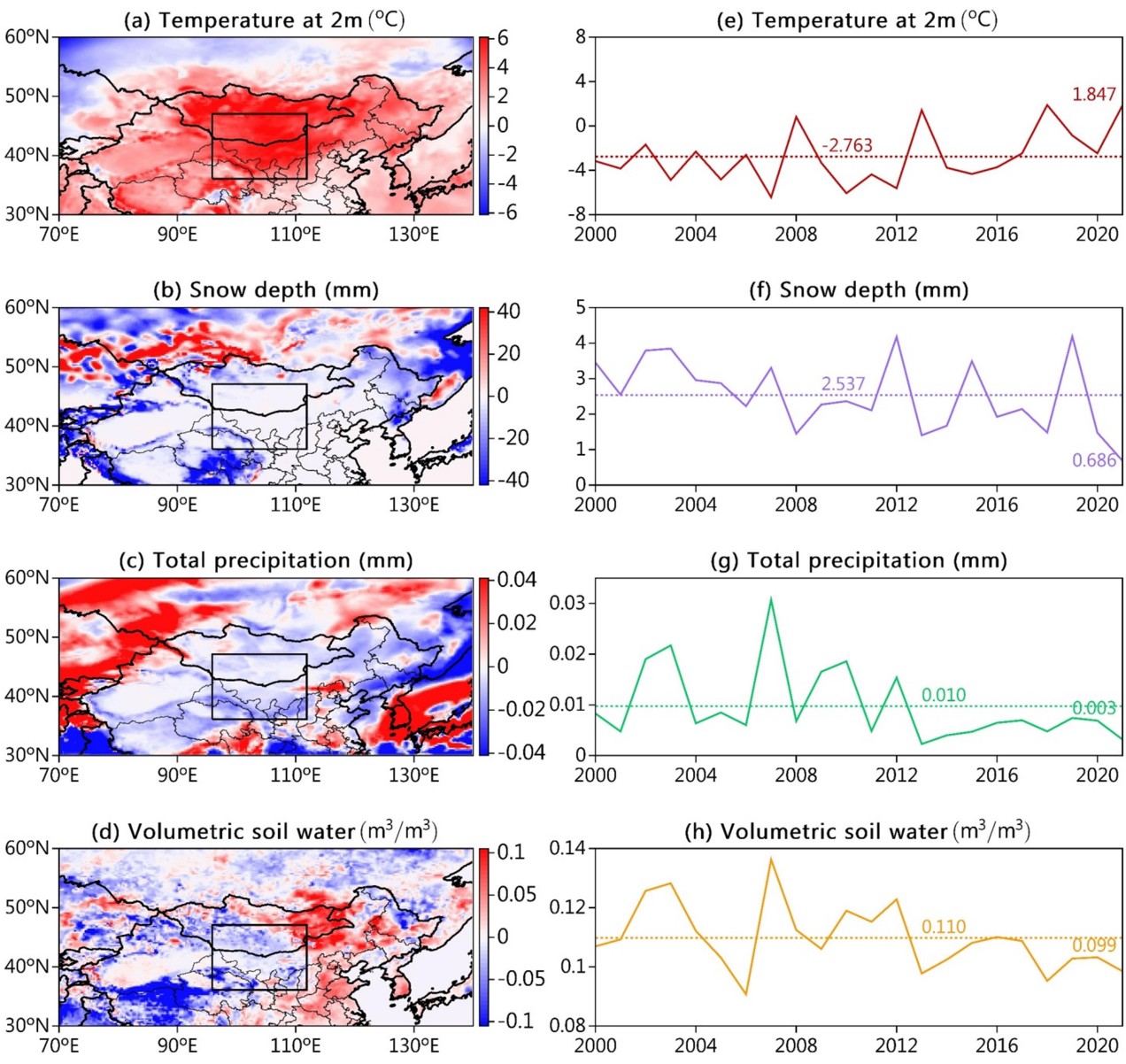

**Figure 18: ERA5 meteorological anomalies two weeks (i.e., March 1–14, 2021) before the 3.15 event: (a–d) anomalies of temperature at 2 m (°C), and snow depth (mm), total precipitation (mm) and volumetric soil water ($m^3/m^3$) with reference to the 2000–2020 climatology. (e–h) Time series of ERA5 meteorological factors two weeks before the 3.15 event averaged over the GD [black box in (a–d): 36°–47°N, 96°–112°W]. The numbers and dashed lines represent the multi-year averages and their locations, respectively. Also, the magnitude for 2021 is labelled.**

