# Peer review of "Record-breaking dust loading during two mega dust storm events over northern China in March 2021: aerosol optical/radiative properties and meteorological drivers"

_Atmospheric Chemistry and Physics, 2021_

## Author Comment (AC1)

The authors are grateful to the referee for your interest and quick comments on our work. We noticed that the referee seemed a little confused regarding the novelty of our work. Therefore, below we will provide a quick response to these confusing questions from the referee.

It is important to clarify that the strong SDS events that occurred in northern China in March 2021 (especially the "3.15" event) attracted widespread concern due to their historical level of intensity and the huge impact on people's lives and the environment. Therefore, such a significant extreme weather event is bound to attract a large number of researchers to conduct in-depth studies on it through different approaches and perspectives. It is for this reason that the referee felt that the main subject of the presented study may overlap with some of the currently published work, and in this way considered our work to be uninnovative. We would like to say that we cannot fully agree with these points of the referee. Firstly, the major difference between this study and other studies (including but not limited to the three works mentioned by the referee) is that we focus on both the "3.15" and "3.27" events, while most of the other studies focus on the "3.15" event. Although the intensity of the "3.27" event was low compared to that of "3.15" event, it was still of historical magnitude, as our study found the maximum daily mean $PM_{10}$ concentration of 2670 μg m$^{-3}$). In our work, we focused on the similarities and differences in the dust transport processes, meteorological causes and impacts between these two SDS events. Secondly, there are no studies to confirm the historical ranking of these two SDS events (are they the strongest in the last 10 years or 20 years as claimed by most studies and reports or not?) by long-term dust aerosol loading observations. To quantify the historical intensity of the contemporaneous dust aerosol loading during March and even during these two SDS event periods, we retrieved a high-resolution DOD dataset from 2000 to the present using MODIS Level 2 products for both Terra and Aqua satellites, which gives us the opportunity to place the intensity of these two SDSs, with DOD as an indicator, in the context of the last two decades to assess their historical ranking, which is one of the greatest highlights of the study. But unfortunately our efforts here were overlooked by the referee. Thirdly, as for the referee's reference to some overlap between this study and three other published papers, we would say that these efforts share some of the similarities but also differ in many regards that are grossly summarized as follows:

"As Section 3.3 of this study, the atmospheric circulation conditions, that triggered the exceptional 3.15 SDS (the strong Mongolian cyclone including its day-by-day movement) are in detail explained in Filonchyk (2022) (their Section 3.1 using ERA5 data))."

**Response:** Firstly, the biggest difference is the different periods they investigated: "3.15" event (Filonchyk (2022)) and the combined "3.15" and "3.27" events (our study). Secondly, there is the difference in the meteorological elements of concern. In Filonchyk (2022), they focus on the day-by-day evolution of geopotential height at 850hpa and the 10m wind field during the "3.15" event, but do not present the joint influence of temperature and sea-level pressure field, and also do not show the hour-by-hour evolution of the sea level pressure field at the beginning of the "3.15" event (14 March). These meteorological factors are key to driving dust emissions, transport and deposition. In addition, Filonchyk (2022) does not

seem to cover the dynamic changes of dust aerosols with the meteorological field during the southward movement of dust plumes, but mainly focuses on the analysis of large-scale synoptic conditions. This may ignore some details of the dynamic evolution of dust aerosols under the influence of meteorological conditions during the dust transport process. For example, Lines 193-194 in our study: *"On March 17, under the influence of southeasterly winds blowing from the Yellow Sea, the dust plume stopped continuing southwards and began to reflux and gradually deposit, and the intensity of its influence weakened significantly compared with the previous two days."*

2. Also, like the presented Section 3.1, the spatiotemporal evolution of the SDS using ground-based along with satellite-based measurements is covered by Filonchyk (2022) (their Section 3.2 and 3.3) and Liang et al. (2022) (their Figure S3, using ERA5 data).
**Response:** Once again, our study time period differs from these two published studies mentioned by the referee. Specifically, Filonchyk (2022) presents the temporal evolution of surface $PM_{10}$ concentrations at several sites from northwest to north China, but misses the impact of dust aerosol transport on northeast China during the "3.27" event (especially on March 28). In Liang (2022), they present the spatial distribution of $PM_{10}$ for 4 specific hours in 2 days, but the dynamic results of the later period of the "3.15" SDS event are not shown, especially the dust backflow phenomenon during the southward movement of the dust plumes on March 16-17. Regarding the satellite observations, Filonchyk (2022) presents the dynamic evolution of the MAIAC AOD, while we present the evolution of the DOD inferred from the MODIS L2 AOD product. DOD, as one of the key parameters for characterizing the optical properties of dust aerosols, describing the columnar optical depth due to the extinction by mineral dust particles. In our study, the use of DOD enables a more realistic characterization of the evolution of dust aerosol loading, whereas AOD cannot distinguish between the contributions of anthropogenic aerosols and dust aerosols, especially those transported over long distances from dust source areas to typical anthropogenic aerosol polluted areas (e.g., the North China Plain).

3. The sources of the dust aeolian aerosols have also been demonstrated using model simulations (Filonchyk (2022)) and HYSPLIT backward trajectories (Liang et al. (2022)).
**Response:** In Filonchyk (2022), they used the BSC-DREAM8b model to study the transport and settlement of desert dust. However, we note that in their study, only the predicted day-by-day spatial distribution of dust concentrations is presented (their Fig. 7), but the evolution of daily emissions of dust aerosols is not characterized, so this makes it difficult to determine the source location of the dust. In addition, we noted that their study only analyzed the model predictions, and the accuracy of model predictions decreases over time. In contrast, aerosol reanalysis (here MERRA-2) can provide a more reasonable and accurate product of dust emissions due to the assimilation of a large number of real-time observations (e.g. MODIS). By comparing with satellite and ground-based observations, Yao et al., (2020) demonstrated the ability of MERRA-2 in characterizing the three-dimensional evolution of dust aerosols during an extreme SDS event in East Asia.

Although backward trajectory analysis can be one of the effective methods to identify dust aerosol sources, it does not clarify the specific location of dust source areas, much less

quantitatively assess dust aerosol emissions. In our study, the analysis of MERRA-2 dust emissions, wind fields, and the results of satellite observations (including DOD and AAI) can provide a more accurate identification of the specific locations of dust source areas.

Yao, W., Che, H., Gui, K., Wang, Y. and Zhang, X.: Can MERRA-2 Reanalysis Data Reproduce the Three-Dimensional Evolution Characteristics of a Typical Dust Process in East Asia? A Case Study of the Dust Event in May 2017, Remote Sens., doi:10.3390/rs12060902, 2020.

4. In addition, as here, the vertical distribution of dust aerosols has also been investigated by CALIPSO retrievals in both Filonchyk (2022), and Liang et al. (2022) works.
**Response:** We used a completely different CALIOP data product to explore the aerosol vertical distribution characteristics. In Filonchyk (2022) and Liang et al. (2022), the CALIOP VFM product was used, while the aerosol profile data product (05kmAPro, V4.21) was used in our study. In these two studies, they aimed to identify dust aerosols from the aerosol type classification products (i.e., VFM), while our study focused on studying the intensity of dust aerosol extinction. The aerosol extinction coefficient can be used as one of the indicators to quantify the dust concentration, and it can provide some observational constraints for the subsequent modeling studies. To summarize, although we all used products from CALIOP, there was a clear difference in focus. In addition, CALIOP has a long revisit period and a fixed transit orbit, which limits its ability to obtain time-continuous vertical measurements. Therefore, to track the dynamic effects of long-range transport of dust aerosols to downstream regions, we additionally introduced ground-based Lidar observations. These observations are not currently covered by other studies during these two strong SDS events, and this should not be overlooked by the referee.

5. Furthermore, like the presented Section 3.5, the climate conditions and anomalies across the study area that could trigger the 3.15 SDS (and not only) event have also been investigated by Zhicong et al. (2021).
**Response:** We would like to say that the periods studied are completely different. Specifically, in our study, we examine the period 2000-2021, which overlaps with MODIS observations, and we specifically focus on the anomalies of four key meteorological factors (Tem, SD, PPT, VSM) in the two weeks preceding these two strong SDS events. Our aim is to characterize the anomalies of four local meteorological factors that are closely related to dust emissions to explain the enhanced dust emissions. In contrast, Yin (2021) focuses on a longer time span (1 December to 31 March) on a longer time scale (1979-2021) or a shorter time scales (2011-2020). In their study, they aimed to emphasize the sub-seasonal variability of the dust source area and to further reveal the joint forces from preceding climate factors, such as sea ice, Nino3.4 index, SST. Although the time periods of the two studies were different, the conclusions obtained were supported by each other.

Finally, we would like to thank the referee once again for their careful comments, and we will highlight the innovative points of this study by clarifying as much as possible the differences between this study and these published studies in the subsequent revised version.

In addition, we will also include an analysis of the effect of dust aerosols on the incoming solar radiation in the revised version, as suggested by the referee. We will also respond to the specific comments mentioned by the referee in a subsequent response.

---

## Author Response (AR1)

Thanks very much for the time and efforts that you have put into reviewing the previous version of the manuscript. We really appreciate all your comments and suggestions that have enabled us to improve the manuscript. The following is a point-to-point response to the reviewer's comments. We have studied comments carefully and have made correction which we hope meet with approval. Revised portion are marked in red in the revised paper.

**Reviewer #1:**

For more than 10 years there was an absence of dust storms across North China. This study applies multiple satellite retrievals, in-situ observations, and reanalysis measurements to characterize two unexpected sand and dust storm (SDS) events in March 2021, formed across northern China. The investigation of those two SDS events is particularly valuable in many ways. More specifically, the presented work investigates the synoptic factors that favor that kind of extreme SDS and highlights the way that different measurements can be combined to interpret the dust transport and emission. The recorded high dust loads originated by the dust storms are also interpreted by calculating the 20-year climatological DOD values across the study area in March. Finally, great effort has been made for interpreting the possible meteorological anomalies that triggered this extreme dust emission. Overall, the manuscript is well-written and well-structured, and the quality of language and visuals are in general satisfactory.

**Response:** Thank you for your positive comments on our work. We have revised it in accordance with your comments or suggestions. For detailed revisions, please refer to the following sections.

**General comments:**

1. This is a reference work of how to treat data and make a thorough study on the way that meteorological (wind speed, temperature etc.) and geophysical factors (soil moisture) can regulate SDS events, receiving information from different sources, and it would be a perfect example for future works in the field. However, I am very confused regarding the novelty of this work. Looking in the already published literature in the field, I have found some works that have been already published, covering the main subject of the presented study. Let me be more concrete.

**Response:** Previously we have provided a quick response to the reviewers' comments to explain the initial idea of this work. However, we regret that the first draft conducted some redundant analyses (some overlapping with existent studies) or similar data were used. After combining your suggestions and those from the second reviewer, we have revised the principal objective of this study, and also made substantial adjustments to the main structure of this study, with the aim of differentiating it from previous studies and also to highlight the innovative aspects of this study. The specific modifications are as follows:

1) The title has been changed from "Two mega sand and dust storm events over northern China in March 2021: transport processes, historical ranking and meteorological drivers" to "*Record-breaking dust loading during two mega dust storm events over northern China in March 2021: aerosol optical/radiative properties and meteorological drivers*".

2) The major difference from the first draft is that the revised draft introduces continuous observations from a sun photometer located in the Beijing area to characterize the aerosol optical, microphysical, and radiative properties during these two SDS events. Two new sections have been introduced, including "Aerosol Optical and Microphysical Properties" in Section 3.2 and "Direct Aerosol Radiative Forcing" in Section 3.3. Please refer to the revised manuscript for details of these changes that were made.

3) The section 3.1 in the initial manuscript has been rewritten by using multi-source satellite fusion AOD, the Himawari-8 dust RGB composite images and corrected visibility observations that were not covered in several existing studies. As a result, the $PM_{10}$ observations, individual MODIS AODs, Suomi NPP VIIRS Imagery and OMPS UVAI used in the initial manuscript have been removed from the revised manuscript.

4) In the revised manuscript we sufficiently discussed the similarities and differences with published related studies to highlight the innovative aspects of this study.

2. The recently published studies of Filonchyk (2022) and Liang et al. (2022) analyze similar datasets (in-situ, remote sensing and models) with the current study, focusing on 3.15 SDS event. "As Section 3.3 of this study, the atmospheric circulation conditions, that triggered the exceptional 3.15 SDS (the strong Mongolian cyclone including its day-by-day movement) are in detail explained in Filonchyk (2022) (their Section 3.1 using ERA5 data)). "
**Response:** Please see the previous quick response.

3. Also, like the presented Section 3.1, the spatiotemporal evolution of the SDS using ground-based along with satellite-based measurements is covered by Filonchyk (2022) (their Section 3.2 and 3.3) and Liang et al. (2022) (their Figure S3, using ERA5 data).
**Response:** We thank the reviewers for their professional advice. First, please see the previous quick response. We apologize for using a similar dataset to the previous study for this study. To address this issue, in the revised manuscript, we removed the datasets (including $PM_{10}$ observations, individual MODIS AODs, Suomi NPP VIIRS Imagery and OMPS UVAI) already used in the previous studies and added some new data sources (including multi-source satellite fusion AOD, Himawari-8 dust RGB composite images and corrected visibility observations) that have not yet been covered. Moreover, continuous observations and retrievals of the optical, microphysical and radiative properties of aerosols during these two SDS events were performed using a sun photometer located in the Beijing area.

These newly added data sources are described as follows:
*Lines 120-128:*
*"2.1.1 Combined AOD from MODIS and VIIRS*
*Aerosol optical depth (AOD) is the column-integrated light extinction by aerosol particles. In this study, Level 2 daily AOD at 550 nm retrieved from Moderate Resolution Imaging Spectroradiometer (MODIS) on board the Terra and Aqua satellites and from Visible*

[revised manuscript text omitted]

*The direct aerosol radiative forcing (DARF in W $m^{-2}$) was calculated by the radiative transfer module similar to the inversion of AERONET under the assumption of cloud-free conditions (García et al., 2008, 2012). The DARF at the top of the atmosphere (TOA) and the Earth's surface (bottom of the atmosphere, BOA) was defined as the difference in the shortwave radiative fluxes with and without aerosol effects in Eqs. (4) and (5) as follows:*

$$DARF_{TOA} = F_{TOA}^{\uparrow 0} - F_{TOA}^{\uparrow} \qquad (4)$$
$$DARF_{BOA} = F_{BOA}^{\downarrow} - F_{BOA}^{\downarrow 0} \qquad (5)$$

*where $F_{TOA}^{\uparrow 0}$ and $F_{BOA}^{\downarrow 0}$ denote the broadband fluxes with no aerosols at TOA and BOA, respectively. $DARF_{TOA}$ is the reflection of solar radiation by aerosols back to space, while $DARF_{BOA}$ indicates the combined effects of absorption and scattering of solar radiation by aerosols. The findings of García et al. (2008) show that the error for the observed solar radiation at the surface on a global scale was +2.1 ± 3.0% for an overestimation of about +9 ± 12 W $m^{-2}$. Subsequently, the DARF at atmosphere ($DARF_{ATM}$) was defined as the difference between $DARF_{BOA}$ and $DARE_{TOA}$ as follow:*

$$DARF_{ATM} = DARF_{TOA} - DARF_{BOA} \qquad (6)$$

*Defined this way, a negative value for DARF indicates aerosol cooling effects, while positive values imply warming.*

**2.2.2. Horizontal visibility**

*Horizontal visibility is closely related to air quality and can be an important indicator of the quality of the atmospheric environment in most scenarios (Gui et al., 2021). In this study, the hourly visibility observations from ~1600 national surface meteorological observation stations across NC provided from the China Meteorological Administration (CMA) during the two SDS events were used to characterize the impacts of the dust plume on air quality. In order to minimize the effect of relative humidity (RH) on horizontal visibility, RH values > 40 and < 99% were converted to the equivalent visibility in dry conditions (i.e., RH < 40%). The correction formula is expressed as VIS/VIS(dry) = 0.26 + 0.4285 $log_{10}$ (100 – RH) (Rosenfeld et al., 2007). The hourly visibility data accompanied by the presence of fog or precipitation were excluded from this study."*

3. The sources of the dust aeolian aerosols have also been demonstrated using model simulations (Filonchyk (2022)) and HYSPLIT backward trajectories (Liang et al. (2022)).

**Response:** First, please see the previous quick response. Moreover, in the revised manuscript, we introduced the dust RGB composite images from Himawari-8 to monitor the dust source at high temporal resolution (see Fig. 2 and Fig. S6). The dust RGB composite images are able to provide a more spatially continuous evolution of the dust plume than the satellite inversion of aerosol-related variables, as it does not need to rely on various retrieval assumptions.

[Figure]

*Figure 2:    Evolution of dust plumes (magenta) as revealed by Himawari-8 dust RGB composite images at 13:00 CST during (a–f) the 3.15 SDS event (March 15–20, 2021) and (g–i) the 3.27 SDS event (March 27–29, 2021), respectively. Overlaid on the RGB imagery is the ERA5 daily mean wind vectors at 10m.*

[Figure]

*Figure S6:    The 3-h evolution of dust plumes (magenta) as revealed by Himawari-8 dust RGB composite images on March 14, 2021. Overlaid on the RGB imagery is the 3-h ERA5 wind vectors at 10m.*

4. In addition, as here, the vertical distribution of dust aerosols has also been investigated by CALIPSO retrievals in both Filonchyk (2022), and Liang et al. (2022) works.

**Response:** First, please see the previous quick response. Moreover, in the revised

manuscript, we separated the contribution of dust from the total extinction coefficient (Fig. 8), which is very different from the previous studies (e.g., Filonchyk et al., 2021; Liang et al., 2021) that focused only on the classification or identification of dust types. In general, we cannot get information such as dust aerosol loading intensity from dust aerosol type classification, instead dust extinction coefficients can do it.

Lines 179-200: *"To characterize the vertical profile of the dust plume, the Level 2 daily 532 nm aerosol profile product (05kmAPro, V4.21) that contains aerosol depolarization, backscatter, and extinction profile from CALIOP/CALIPSO (Winker et al., 2010) was used. We use the methodology in Yu et al. (2015) to derive the dust extinction profile. To reduce uncertainty, only high-quality extinction profile data with a CAD (cloud aerosol discrimination) score of between −100 and −90 were used. The aerosol profile product also provides an extinction quality control flag (Ext_QC) to indicate problematic retrievals. This study only uses layers with Ext_QC values of 0, 1, 18, and 16 (Winker et al., 2013). For each aerosol backscatter coefficient profile, we infer the ratio of dust to total backscatter ($f_d$) at each altitude from the following equation:*

$$f_d = \frac{(\delta - \delta_{nd})(1+\delta_d)}{(\delta_d - \delta_{nd})(1+\delta)} \tag{2}$$

*where $\delta$ is CALIOP observed particulate depolarization ratio, $\delta_d$ and $\delta_{nd}$ are a priori knowledge of depolarization ratios of dust and non-dust aerosols respectively. To account for various types of non-dust aerosols with different depolarization ratio and for the variability of dust shape and size, we follow Song et al. (2021) and use the $f_d$ that was based on the mean of the lowest ($\delta_d = 0.30$ and $\delta_{nd} = 0.07$) and the highest ($\delta_d = 0.20$ and $\delta_{nd} = 0.02$) dust scenario. By assuming a dust lidar ratio (LR) (i.e., extinction-to-backscatter ratio) of 40 sr at 532 nm (Yu et al., 2015), we derive dust extinction coefficient (DEC) profile from dust backscatter coefficient."*

[Figure]

*Figure 8: CALIOP/CALIPSO snapshots of dust plumes for (a) March 15, (b) March 16, (c) March 27, and (d)*

*March 28, 2021. The first column presents the CALIPSO tracks (red or blue lines) and the second column shows the 532 nm dust extinction coefficients (km$^{-1}$). Surface elevation is indicated by the gray filled line.*

5. Furthermore, like the presented Section 3.5, the climate conditions and anomalies across the study area that could trigger the 3.15 SDS (and not only) event have also been investigated by Zhicong et al. (2021).

**Response:** Please see the previous quick response.

6. The Authors need to clarify the way that this study contributes meaningfully to the existing published literature. Therefore, in my point of view, this analysis does not expand or fill the gaps of the recently published works and my recommendation is to reject the paper in this form. However, I believe the paper could eventually be published but significant supplement analysis must be added to the revised manuscript. For instance, the effect of these sudden SDS events, including a huge amount of dust burden, on the incoming solar radiation and thus Earth's climate could be a very interesting topic for further investigation. If you decide to proceed in this manuscript form it is necessary to define the contribution of your work.

**Response:** We thank the reviewers for their professional advice. In the revised manuscript, we have changed the principal objective of this study to emphasize aerosol optical, microphysical, and radiative properties during two mega SDS events rather than dust transport processes as in the initial manuscript, which is considered to be partially overlapping with several existing studies. In the revised manuscript, we introduced the principal objective of this study by comparing it with other similar studies, and these sentences are introduced below to emphasize this points.

In the introduction section: *"To date, several studies (e.g., Liang et al., 2021; Filonchyk, 2022; Filonchyk and Peterson, 2022) have been conducted to characterize the severe SDS event in March 2021. Most of these studies have focused on investigating the evolution and transport processes of the dust plume during the 3.15 event and assess its impact on the air quality by using particulate matter (PM$_{10}$) concentration observations and individual satellite retrieval products. However, few studies have been carried out on the optical, microphysical, and radiative properties of aerosols during the March 2021 SDS events, which are critical to accurately assess the weather and climate effects associated with enhanced dust loadings. Furthermore, these existing studies focus on the 3.15 event, and the 3.27 event, which also has a huge impact, has not received sufficient attention. Therefore, it is essential to combine the two events to elucidate their similarities and differences in terms of dust sources, aerosol optical, microphysical, and radiative properties, and meteorological drivers."*

We have included in the revised version, as suggested by the reviewer, an analysis of the effect of dust aerosols on the incoming solar radiation based on continuous observations and retrievals from a sun photometer located in the Beijing area, along with the evolution of the optical and microphysical properties of aerosols under the influence of dust plumes during the two SDS events (see Table1, Figs. 5 and 6). Two new sections have been added,

including "*Aerosol Optical and Microphysical Properties*" in Section 3.2 and "*Direct Aerosol Radiative Forcing*" in Section 3.3. Please refer to the revised manuscript for details of these changes that were made. Briefly, the following are some of the valuable conclusions obtained from this study in terms of aerosol optical, microphysical and radiative properties.

Lines 30-35 in Abstract: "*Despite the shorter duration of the 3.27 event relative to the 3.15 event, sun photometer and satellite observations in Beijing recorded a larger peak AOD (~2.5) in the former than in the latter (~2.0), which was mainly attributed to the short-term intrusion of coarse-mode dust particles with larger effective radii (~1.9 μm) and volume concentrations (~2.0 μm³ μm⁻²) during the 3.27 event. The direct aerosol radiative forcing (DARF) induced by dust was estimated to be −92.1 and −111.4 W m⁻² at the top of the atmosphere, −184.7 and −296.2 W m⁻² at the surface, and +92.6 and +184.8 W m⁻² in the atmosphere in Beijing during the 315 and 3.27 event, respectively.*"

*Table 1. Daily arithmetic mean of aerosol optical/microphysics parameters at Beijing-CAMS site during the two SDS events.*

| | 3.15 event | | | 3.27 event | |
|---|---|---|---|---|---|
| Day | Mar. 16 | Mar. 17 | Mar. 20 | Mar. 28 | Mar. 29 |
| $N_{inst}{}^{a}$ | 3 | 3 | 4 | 1 | 1 |
| $SSA_{440nm}{}^{b}$ | 0.984 ± 0.002 | 0.915 ± 0.025 | 0.950 ± 0.042 | 0.900 | 0.986 |
| $AAOD_{440nm}{}^{b}$ | 0.004 ± 0.001 | 0.164 ± 0.047 | 0.016 ± 0.014 | 0.249 | 0.002 |
| $AAE_{440-870nm}{}^{c}$ | 0.849 ± 0.217 | 2.973 ± 0.269 | 1.345 ± 1.078 | 2.799 | 0.843 |
| $R_{eff_t}$ (μm)$^{b}$ | 0.738 ± 0.044 | 0.940 ± 0.061 | 1.305 ± 0.234 | 1.019 | 0.878 |
| $R_{eff_f}$ (μm)$^{b}$ | 0.126 ± 0.010 | 0.157 ± 0.018 | 0.113 ± 0.007 | 0.080 | 0.146 |
| $R_{eff_c}$ (μm)$^{b}$ | 1.727 ± 0.057 | 1.661 ± 0.058 | 2.360 ± 0.200 | 1.850 | 1.722 |
| $Volume_t$ (μm³ μm⁻²)$^{b}$ | 0.136 ± 0.008 | 1.110 ± 0.066 | 0.270 ± 0.053 | 2.084 | 0.081 |
| $Volume_f$ (μm³ μm⁻²)$^{b}$ | 0.014 ± 0.001 | 0.088 ± 0.010 | 0.011 ± 0.002 | 0.076 | 0.007 |
| $Volume_c$ (μm³ μm⁻²)$^{b}$ | 0.121 ± 0.007 | 1.022 ± 0.056 | 0.259 ± 0.054 | 2.007 | 0.074 |

$^{a}$ Number of instantaneous observations. $^{b}$ Optical parameters at a wavelength of 440 nm. $^{c}$ Absorption angström exponent between 440 and 870 nm.

[Figure]

Figure 5: Daily variation of AOD at 550 nm (top row), EAE between 440 and 870 nm (middle row), and FMF

*(bottom row) at Beijing-CAMS site during (a–c) the 3.15 event and (d-e) the 3.27 event. The instantaneous AOD values from the MODIS and VIIRS sensors, which were derived using the Dark Target (DT) and Deep Blue (DB) algorithms, respectively, are given in the top panel.*

[Figure]

*Figure 6: Daily variation of the aerosol volume-size distributions at Beijing-CAMS site during the two SDS events.*

In addition, based on the previous studies, we fully justify or discuss the results or conclusions of this study. For example:

Lines 355-345: "*CMA station records (Fig.3) show that, during the 3.15 event, horizontal visibility first reached a minimum on March 15 in most of NC, including Gansu, southwestern Inner Mongolia, Ningxia, northern Shaanxi, northern Shanxi, Hebei and Beijing, with the number of stations with daily mean visibility below 500 m reaching 19, due to dust plume deposition. Such remarkable contribution of dust aerosols to air quality is also supported by the results revealed by Filonchyk (2022) using $PM_{10}$ observations. Influenced by SDS, $PM_{10}$ concentrations in some regions of China were found to exceed 7000.0 $\mu g\ m^{-3}$ on March 15. Similarly, Liang et al. (2021) claimed that the 3.15 SDS event was the most severe in China in the past decade, with $PM_{10}$ concentration in Beijing reaching 6450 $\mu g\ m^{-3}$. Our results show that on March 15, the instantaneous surface horizontal visibility was below 50m near the lower limit of the monitoring threshold at seven sites (Fig. S2).*"

[Figure]

*Figure 3:    Evolution of observed daily mean (presented as averages close to the MODIS and VIIRS observation time range, i.e., approximately 10:00 to 14:00 CST) corrected visibility during (a–f) the 3.15 SDS event (March 15–20, 2021) and (g–i) the 3.27 SDS event (March 27–29, 2021), respectively.*

[Figure]

*Figure S2:    Evolution of observed daily minimum hourly corrected visibility during (a–f) the 3.15 event (March 15–20, 2021) and (g–i) the 3.27 event (March 27–29, 2021).*

Lines 369-373: "*Here, multi-satellite fusion provides more available retrievals and regional details by incorporating the observed dust loading at different satellite transit times than the individual satellite data sources used in the previous similarity studies (e.g., Filonchyk, 2022; Filonchyk and Peterson, 2022).*"

Lines 469-472: *"Generally, the instantaneous DARF on March 28 estimated in this study was stronger than similar studies previously performed during several strong SDS events, such as in Beijing in May 2017 (Filonchyk et al., 2021) and over the Indo-Gangetic Basin in May 2018 (Tiwari et al., 2019). "*

*Lines 498-451: "These findings are consistent with the results of Jin et al. (2022) using inverse modelling and Liang et al. (2021) using backward trajectory simulations. They revealed that wind-blown dust emissions originated from both China and Mongolia contribute to the SDS events that occur in spring 2021."*


*For this purpose, we use the AERONET Version 3 spectral deconvolution algorithm (SDA) daily products (Level 2.0 data). Given that AERONET SDA only provide the AODc at 500 nm, the AERONET AODc is converted to 550 nm in this study using the Ångström exponent to compare with MODIS DOD retrievals. For March dust retrievals, between 2000 and 2021, there are 121 MODIS daily mean DOD retrievals collocated with 7 AERONET sites located within NC (Fig. S1). Results showed that the MODIS DOD is in good agreement with AERONET AODc (Person correlation coefficient = 0.82), although the former generally overestimated latter in NC (root-mean-square error = 0.28)."*

[Figure]

*Figure S1: Scatter plot of the daily mean MODIS DOD against the AERONET coarse-mode AOD (AODc) retrieved at 550nm. The 1-to-1 line and linear regression line are shown by black dotted and red solid lines, respectively. The number of sites (Sites), matchups (N), Pearson correlation coefficient (R), slope, and root mean square error (RMSE) of the linear regression are indicated in the lower right of the panel.*

2. Line 140: please change "ERA5 and MERRA-2 reanalyses" to "Reanalysis datasets".
**Response:** Corrected.

3. Line 148: please remove the dash in "10-m".

**Response:** Corrected.

4. Line 217: the value of AEC is missing in the parenthesis.

**Response:** Corrected. In addition, "AEC" has been changed to "DEC".

5. In Section 3.1, five (three satellite-based and two ground-based) datasets are used to characterize the two SDS events. The quantitative contribution of each satellite-based dataset against in-situ measurements should be included in this section. What's extra information gives the UVAI against MODIS-DOD? Again, UVAI and MODIS-DODs against $PM_{10}$? If not, authors should choose the most appropriate datasets to reduce the redundant information.

**Response:** The section 3.1 in the initial manuscript has been rewritten by using multi-source satellite fusion AOD, the Himawari-8 dust RGB composite images and corrected visibility observations that were not covered in several existing studies. As a result, the $PM_{10}$ observations, individual MODIS AODs, Suomi NPP VIIRS Imagery and OMPS UVAI used in the initial manuscript have been removed from the revised manuscript.

6. In Figs. 2, 4 etc. the latitude and longitude information on the y-axis and x-axis respectively is missing. For instance, in line 235 you mentioned: "(confined within 40-50 ºN)" but there is no such information in the graph. Please add this information.

**Response:** Corrected.

7. On Mar-28, can you please explain the differences between DOD and $PM_{10}$ values in Liaoning (122 ºE, 42 ºN). DOD documents notably high values (>2.5) while $PM_{10}$ lies within $100-400 \ \mu g/m^3$.

**Response:** In the revised manuscript, the data sources and descriptions about $PM_{10}$ have been removed. Nevertheless, the difference in magnitude between DOD and $PM_{10}$ is mainly attributed to the difference between ground-based and satellite observation times, and influenced by the vertical distribution of dust aerosols transported to the Liaoning region. Specifically, the $PM_{10}$ map is the daily average concentration, while the DOD map retrieved by MODIS is the average of the transit times of the two satellites Terra and Aqua. DOD is a physical quantity that represents the columnar dust loading, while $PM_{10}$ characterizes the near-surface dust concentration.

8. Lines 252−254: Please rephrase, to show that the peak in UVAI is documented near the Bohai Sea on Mar-20.

**Response:** In the revision, the redundant UVAI analysis has been removed.

9. Wind magnitudes and directions between Fig. 2 and Fig. 5 are unanticipated different. Please check again. What's the correct underlying meteorology?

**Response:** This may be a misunderstanding by the reviewer. This difference is due to the wind fields presented in Figs. 2 and 5 at completely different heights, the former at 10 m and the latter at 850 hpa.

10. Line 362: I think you mean 2000−2020 instead of 2001−

**Response:** Corrected.

11. Lines 382−383: Please add a reference here.

**Response:** The following two references were included.
1) Che, H., Gui, K., Xia, X., Wang, Y., Holben, B. N., Goloub, P., Cuevas-Agulló, E., Wang, H., Zheng, Y., Zhao, H. and Zhang, X.: Large contribution of meteorological factors to inter-decadal changes in regional aerosol optical depth, Atmos. Chem. Phys., 19, 10497–10523, doi:10.5194/acp-19-10497-2019, 2019.
2) Pu, B. and Jin, Q.: A record-breaking trans-Atlantic African dust plume associated with atmospheric circulation extremes in June 2020, Bull. Am. Meteorol. Soc., 1–41, doi:10.1175/bams-d-21-0014.1, 2021.

12. Lines 392−399: What's extra information gives this paragraph to the already existing analysis?

**Response:** The purpose of this part is to evaluate the ranking of the regional-averaged dust loading during these two mega SDS events occurred in March 2021 on different time scales for a more profound portrayal of the extremity of these two events. In the revised manuscript, we have moved Figs 13 and 14 to the supplementary information to reduce the similar presentation.

13. Line 410: Delete "S" from "Fig. S15".

**Response:** Corrected.

**Reviewer #2:**

The paper "Two mega sand and dust storm events over northern China in March 2021: transport processes, historical ranking and meteorological drivers" by Ke Gui et al. investigates two remarkable sand and dust storm (SDSs) occurred on March 15–20, 2021 and March 27–29, 2021. The study characterizes the origins, transport processes, magnitudes of impact, and meteorological causes of these two SDS events, through satellite and ground-based observations combined with atmospheric reanalysis data. The study falls within the scope of ACP. The manuscript is well-written/structured, the presentation clear, and the language fluent. However, the submitted study is subject to major deficiencies in principal ACP evaluation criteria. Here are some of my main comments which I think will help the authors to improve their manuscript.

**Response:** We thank the reviewers for their professional advice. In the revised manuscript, we have changed the principal objective of this paper to emphasize aerosol optical, microphysical, and radiative properties during two mega SDS events rather than dust transport processes as in the initial manuscript, which is considered to be partially overlapping with several existing studies. After combining your suggestions and those from the first reviewer, we have made numerous improvements to the structure and content of this study, mainly including:

1) The title has been changed from "Two mega sand and dust storm events over northern China in March 2021: transport processes, historical ranking and meteorological drivers" to "*Record-breaking dust loading during two mega dust storm events over northern China in March 2021: aerosol optical/radiative properties and meteorological drivers*".

2) The major difference from the first draft is that the revised draft introduces continuous observations from a sun photometer located in the Beijing area to characterize the aerosol optical, microphysical, and radiative properties during these two SDS events. Two new sections have been introduced, including "Aerosol Optical and Microphysical Properties" in Section 3.2 and "Direct Aerosol Radiative Forcing" in Section 3.3. Please refer to the revised manuscript for details of these changes that were made.

3) The section 3.1 in the initial manuscript has been rewritten by using multi-source satellite fusion AOD, the Himawari-8 dust RGB composite images and corrected visibility observations that were not covered in several existing studies. As a result, the $PM_{10}$ observations, individual MODIS AODs, Suomi NPP VIIRS Imagery and OMPS UVAI used in the initial manuscript have been removed from the revised manuscript.

4) In the revised manuscript we sufficiently discussed the similarities and differences with published related studies to highlight the innovative aspects of this study.

For detailed revisions, please refer to the following sections.

(1) The submitted manuscript in general presents limited novel concepts, ideas, tools, or data. An exception is the historical ranking of the dust events, although this is a secondary

objective of the manuscript. To be more specific, as stated by the authors, the manuscript's principal objective is to "characterize the origins, transport processes, magnitudes of impact, and meteorological causes of these two SDS events". With respect to the most significant of the two SDSs events discussed here, the event on March 15–20, the meteorology and impact of the 3.15 SDS is discussed in Filonchyk et al., 2022 "Characteristics of the severe March 2021 Gobi Desert dust storm and its impact on air pollution in China", while the transport processes are provided by Liang et al. (2021) "Revealing the dust transport processes of the 2021 mega dust storm event in northern China". Thus, "the origins, transport processes, magnitudes of impact, and meteorological causes" have already been discussed. A recommendation would be to focus on the second event of March 27–29, 2021 which has not been discussed so far and on the historical ranking of the dust event, which however, according to manuscript and extend of material, is interpreted as a secondary objective.

**Response:** We thank the reviewers for their professional advice. We regret that this study conducted some redundant analyses (some overlapping with existent studies) or similar data were used. After a careful reading of the literature mentioned by the reviewers, including but not limited to these two papers (i.e., Filonchyk et al., 2022 and Liang et al., 2021), we have revised the principal objective of this study, and also made substantial adjustments to the main structure of this study, with the aim of differentiating it from previous studies and also to highlight the innovative aspects of this study. The specific modifications are as follows:

1) The title has been changed to "*Record-breaking dust loading during two mega dust storm events over northern China in March 2021: aerosol optical/radiative properties and meteorological drivers*".

2) In the introduction section, the purpose of our study is clearly presented as follows:
"*To date, several studies (e.g., Liang et al., 2021; Filonchyk, 2022; Filonchyk and Peterson, 2022) have been conducted to characterize the severe SDS event in March 2021. Most of these studies have focused on investigating the evolution and transport processes of the dust plume during the 3.15 event and assess its impact on the air quality by using particulate matter ($PM_{10}$) concentration observations and individual satellite retrieval products. However, few studies have been carried out on the optical, microphysical, and radiative properties of aerosols during the March 2021 SDS events, which are critical to accurately assess the weather and climate effects associated with enhanced dust loadings. Furthermore, these existing studies focus on the 3.15 event, and the 3.27 event, which also has a huge impact, has not received sufficient attention. Therefore, it is essential to combine the two events to elucidate their similarities and differences in terms of dust sources, aerosol optical, microphysical, and radiative properties, and meteorological drivers.*".

3) The parts (i.e., section 3.1 in the initial manuscript) that were considered to overlap with previous studies were completely removed. In the revised manuscript, a new section 3.1 has been introduced with the following main purpose:
"*Several studies have been performed to reveal the transport processes of dust aerosols*

*during the 3.15 event and their impacts on near-surface air quality using PM$_{10}$ concentration as indicator (Liang et al., 2021; Filonchyk, 2022; Filonchyk and Peterson, 2022). However, no studies have focused on the 3.27 event and the differences between the two events have not been explored. This section will provide an overview of these two SDS events based on satellite RGB images, horizontal visibility observations, and multi-satellite fusion to reveal the similarities and differences between them from different perspectives.*"
Please refer to the revised manuscript for details of these changes that were made.

4) To achieve the principal objective of this study (i.e., aerosol optical/radiative properties), we introduced the continuous observations from a sun photometer located in the Beijing area to characterize the aerosol optical, microphysical, and radiative properties during these two SDS events. Two new sections have been introduced, including "*Aerosol Optical and Microphysical Properties*" in Section 3.2 and "*Direct Aerosol Radiative Forcing*" in Section 3.3. Please refer to the revised manuscript for details of these changes that were made. Briefly, the following are some of the valuable conclusions obtained from this study in terms of aerosol optical, microphysical and radiative properties.

Lines 30-35 in Abstract: "*Despite the shorter duration of the 3.27 event relative to the 3.15 event, sun photometer and satellite observations in Beijing recorded a larger peak AOD (~2.5) in the former than in the latter (~2.0), which was mainly attributed to the short-term intrusion of coarse-mode dust particles with larger effective radii (~1.9 μm) and volume concentrations (~2.0 μm$^3$ μm$^{-2}$) during the 3.27 event. The direct aerosol radiative forcing (DARF) induced by dust was estimated to be −92.1 and −111.4 W m$^{-2}$ at the top of the atmosphere, −184.7 and −296.2 W m$^{-2}$ at the surface, and +92.6 and +184.8 W m$^{-2}$ in the atmosphere in Beijing during the 315 and 3.27 event, respectively.*"

(2) Providing the objectives of the study, it is stated by the authors that "although these two studies have strengthened our understanding of the 3.15 mega SDS event in 2021, the sources, three-dimensional evolutionary features during transport processes, historical ranking, and local meteorological anomalies of the 3.15 and 3.27 SDS events have not yet been elucidated." However, with respect to ACP evaluation criteria of "giving proper credit to related work and clearly indicate their own new/original contribution?", the significant published study of Filonchyk et al. (2022), presenting the characteristics of the severe March 2021 Gobi Desert dust storm and its impact on air pollution in China is not mentioned, and the outcomes not compared, nor discussed in terms of discrepancies, similarities – although substantial – and conclusions. A strong recommendation is to extensively discusses the similarities/differences/conclusions of the related published studies, and build on top of the previous studies.
**Response:** We regret that some existing studies have not been sufficiently presented and discussed to distinguish their similarities and differences from the present study. In the revised manuscript, we introduced the objectives of this study by comparing it with other similar studies, and these sentences are reintroduced below to emphasize this point.

In the introduction section: "*To date, several studies (e.g., Liang et al., 2021; Filonchyk, 2022; Filonchyk and Peterson, 2022) have been conducted to characterize the severe SDS event in March 2021. Most of these studies have focused on investigating the evolution and transport processes of the dust plume during the 3.15 event and assess its impact on the air quality by using particulate matter ($PM_{10}$) concentration observations and individual satellite retrieval products. However, few studies have been carried out on the optical, microphysical, and radiative properties of aerosols during the March 2021 SDS events, which are critical to accurately assess the weather and climate effects associated with enhanced dust loadings. Furthermore, these existing studies focus on the 3.15 event, and the 3.27 event, which also has a huge impact, has not received sufficient attention. Therefore, it is essential to combine the two events to elucidate their similarities and differences in terms of dust sources, aerosol optical, microphysical, and radiative properties, and meteorological drivers.*"

In addition, based on the previous studies, we fully justify or discuss the results or conclusions of this study. For example:

Lines 355-345: "*CMA station records show that, during the 3.15 event, horizontal visibility first reached a minimum on March 15 in most of NC, including Gansu, southwestern Inner Mongolia, Ningxia, northern Shaanxi, northern Shanxi, Hebei and Beijing, with the number of stations with daily mean visibility below 500 m reaching 19, due to dust plume deposition. Such remarkable contribution of dust aerosols to air quality is also supported by the results revealed by Filonchyk (2022) using $PM_{10}$ observations. Influenced by SDS, $PM_{10}$ concentrations in some regions of China were found to exceed 7000.0 $\mu g\ m^{-3}$ on March 15. Similarly, Liang et al. (2021) claimed that the 3.15 SDS event was the most severe in China in the past decade, with $PM_{10}$ concentration in Beijing reaching 6450 $\mu g\ m^{-3}$. Our results show that on March 15, the instantaneous surface horizontal visibility was below 50m near the lower limit of the monitoring threshold at seven sites (Fig. S2).*"

Lines 369-373: "*Here, multi-satellite fusion provides more available retrievals and regional details by incorporating the observed dust loading at different satellite transit times than the individual satellite data sources used in the previous similarity studies (e.g., Filonchyk, 2022; Filonchyk and Peterson, 2022).*"

Lines 469-472: "*Generally, the instantaneous DARF on March 28 estimated in this study was stronger than similar studies previously performed during several strong SDS events, such as in Beijing in May 2017 (Filonchyk et al., 2021) and over the Indo-Gangetic Basin in May 2018 (Tiwari et al., 2019).*"

Lines 498-451: "*These findings are consistent with the results of Jin et al. (2022) using inverse modelling and Liang et al. (2021) using backward trajectory simulations. They revealed that wind-blown dust emissions originated from both China and Mongolia contribute to the SDS events that occur in spring 2021.*"

*For this purpose, we use the AERONET Version 3 spectral deconvolution algorithm (SDA) daily products (Level 2.0 data). Given that AERONET SDA only provide the AODc at 500 nm, the AERONET AODc is converted to 550 nm in this study using the Ångström exponent to compare with MODIS DOD retrievals. For March dust retrievals, between 2000 and 2021, there are 121 MODIS daily mean DOD retrievals collocated with 7 AERONET sites located within NC (Fig. S1). Results showed that the MODIS DOD is in good agreement with AERONET AODc (Person correlation coefficient = 0.82), although the former generally overestimated latter in NC (root-mean-square error = 0.28)."*

[Figure]

*Figure S1: Scatter plot of the daily mean MODIS DOD against the AERONET coarse-mode AOD (AODc)*
*retrieved at 550nm. The 1-to-1 line and linear regression line are shown by black dotted and red solid lines,*
*respectively. The number of sites (Sites), matchups (N), Pearson correlation coefficient (R), slope, and root mean*
*square error (RMSE) of the linear regression are indicated in the lower right of the panel.*

(4) The study could be improved in terms of providing and outlining more extensively the information of the scientific methods and assumptions used, which are only briefly discussed, in terms of limitations of the implemented datasets. For instance, very few information on the quality assurance criteria is provided. With respect to CALIPSO, as QA only the CAD score is mentioned that it is implemented, while in the literature, significantly more QA filters are recommended (e.g. Tackett et al., 2018). Another point could be the particulate depolarization ratio, when observed lower to 0.3, indicates mixtures of dust and non-dust components, thus the provided CALIPSO information, considered in the study as pure-dust, overestimates the actual dust extinction coefficient. With respect to MODIS-retrieved DOD, which are the limitations in terms of Deep-Blue algorithm, Ångström exponent, SSA and the quality of the DOD product? Which are the attenuation effects – thus the limitations – of the different datasets (active and passive) and how do they compare in terms of observations? The SDSs are transported detached in extended domains over China, and how does this compare with only the surface-based in-situ? There is a wealth of datasets incorporated in the study, however, the dataset observations are not extensively spatio-temporarily inter-compared to account for and extract the closest representation and actual state of the SDSs.

**Response:** We thank the reviewers for their professional advice.

1) In order to completely separate the contribution of dust to the total aerosol extinction, we used the following method in the revised manuscript. In addition, with reference to similar previous studies, more quality control methods were used.

Lines 179-200: *"To characterize the vertical profile of the dust plume, the Level 2 daily 532 nm aerosol profile product (05kmAPro, V4.21) that contains aerosol depolarization, backscatter, and extinction profile from CALIOP/CALIPSO (Winker et al., 2010) was used. We use the methodology in Yu et al. (2015) to derive the dust extinction profile. To reduce uncertainty, only high-quality extinction profile data with a CAD (cloud aerosol discrimination) score of between −100 and −90 were used. The aerosol profile product also provides an extinction quality control flag (Ext_QC) to indicate problematic retrievals. This study only uses layers with Ext_QC values of 0, 1, 18, and 16 (Winker et al., 2013). For each aerosol backscatter coefficient profile, we infer the ratio of dust to total backscatter ($f_d$) at each altitude from the following equation:*

$$f_d = \frac{(\delta - \delta_{nd})(1 + \delta_d)}{(\delta_d - \delta_{nd})(1 + \delta)} \qquad (2)$$

*where $\delta$ is CALIOP observed particulate depolarization ratio, $\delta_d$ and $\delta_{nd}$ are a priori knowledge of depolarization ratios of dust and non-dust aerosols respectively. To account for various types of non-dust aerosols with different depolarization ratio and for the variability of dust shape and size, we follow Song et al. (2021) and use the $f_d$ that was based*

*on the mean of the lowest ($\delta_d$ = 0.30 and $\delta_{nd}$ = 0.07) and the highest ($\delta_d$ = 0.20 and $\delta_{nd}$ = 0.02) dust scenario. By assuming a dust lidar ratio (LR) (i.e., extinction-to-backscatter ratio) of 40 sr at 532 nm (Yu et al., 2015), we derive dust extinction coefficient (DEC) profile from dust backscatter coefficient."*

2)    Regarding the uncertainty of the MODIS-retrieved DOD mentioned by the reviewer. Admittedly, the uncertainty in each input variable contributes to the uncertainty in the DOD retrievals, but it is difficult for us to distinguish between them. Therefore, to ensure the accuracy of the MODIS-retrieved DOD, we evaluated it comparatively using the inversion results of AERONET with reference to the previous studies (e.g., Pu and Ginoux, 2018; Song et al., 2021). Also, some other uncertainty descriptions were included.

Lines 161-179: *"MODIS dust detection is subject to a number of uncertainties. Over land, the derived MODIS DOD here denotes the coarse-mode (aerodynamic diameters larger than 1 µm) contribution of dust only and does not include its fine-mode contribution. Estimates by Kok et al. (2017) suggest that the exclusion of submicron dust aerosol could induce around 3% underestimation of the global atmospheric dust mass load and around 15% underestimation of the global DOD. Terra and Aqua DOD values at daily and monthly scales have previously been validated with AERONET stations globally (Pu and Ginoux, 2018; Song et al., 2021). Spatially, when comparing the MODIS-derived DOD climatology with the DOD retrieved from the Cloud–Aerosol Lidar with Orthogonal Polarization (CALIOP) aboard the Cloud–Aerosol Lidar and Infrared Pathfinder Satellite Observation (CALIPSO) satellite, the climatological mean of the MODIS DOD generally compares well with CALIOP (Pu and Ginoux, 2018; Song et al., 2021). Here we compare the combined daily MODIS DOD against AERONET stations in NC (Fig. S1). It should be noted that to date, there is no valid method to derive DOD from AERONET AOD measurements. Therefore, we use coarse-mode AOD (AODc) from AERONET measurements as a proxy for DOD (Pu and Ginoux, 2018; Song et al., 2021) to compare with our DOD datasets in March for NC.*

*For this purpose, we use the AERONET Version 3 spectral deconvolution algorithm (SDA) daily products (Level 2.0 data). Given that AERONET SDA only provide the AODc at 500 nm, the AERONET AODc is converted to 550 nm in this study using the Ångström exponent to compare with MODIS DOD retrievals. For March dust retrievals, between 2000 and 2021, there are 121 MODIS daily mean DOD retrievals collocated with 7 AERONET sites located within NC (Fig. S1). Results showed that the MODIS DOD is in good agreement with AERONET AODc (Person correlation coefficient = 0.82), although the former generally overestimated latter in NC (root-mean-square error = 0.28)."*

[Figure]

*Figure S1: Scatter plot of the daily mean MODIS DOD against the AERONET coarse-mode AOD (AODc) retrieved at 550nm. The 1-to-1 line and linear regression line are shown by black dotted and red solid lines, respectively. The number of sites (Sites), matchups (N), Pearson correlation coefficient (R), slope, and root mean square error (RMSE) of the linear regression are indicated in the lower right of the panel.*

3) Regarding the limitations of different data sets

Different datasets, especially satellite retrieval products, have many limitations, such as being influenced by clouds and algorithmic assumptions. Therefore, in order to address these issues, we have introduced some new data sources and presentations in the revised manuscript. First, we used a multi-satellite (including MODIS/Terra, MODIS/Aqua and VIIRS) combined AOD to improve the coverage of the valid daily average AOD values (Fig. 4). Secondly, we introduced the dust RGB composite images from Himawari-8 to monitor the source, genesis and movement of dust at high temporal resolution (Fig.2). The dust RGB composite images are able to provide a more spatially continuous evolution of the dust plume than the satellite inversion of aerosol-related variables, as it does not need to rely on various retrieval assumptions.

[Figure]

*Figure 2:    Evolution of dust plumes (magenta) as revealed by Himawari-8 dust RGB composite images at*

[Figure]

*Figure 4:    Evolution of MODIS and VIIRS combined daily mean AOD during (a–f) the 3.15 event and (g–i) the 3.27 event.*

4)    To illustrate and extract the closest representation and actual state of the SDS, we carried out a spatial-temporal intercomparison between the multi-source satellite-retrieved AODs and ground-based sun photometer observations in Beijing.

Lines 364-369: *"The accuracy of the MODIS and VIIRS instantaneous AOD values retrieved using individual algorithms (i.e., DT and DB algorithms) was confirmed by a synergistic comparison with ground-based sun photometer observations located in Beijing (i.e., Beijing-CAMS site) (Fig. 5). To complete the spatial collocation, the satellite-retrieved AOD values is represented by the average value of all the satellite product pixels within a 25 km radius (Sayer et al., 2013) around the Beijing-CAMS site. The results show that both the MODIS and VIIRS instantaneous AOD values are in high agreement with ground-based observations."*

[Figure]

*Figure 5: Daily variation of AOD at 550 nm (top row), EAE between 440 and 870 nm (middle row), and*

(5) The ground-based lidar observations are a novelty however, in terms of dataset. At this point, the authors could provide more information to characterize the SDSs, including the intensive properties of dust (e.g. LR used), the pre-processing of the dataset and retrieval (e.g. Raman/Klett), the temporal averaging techniques, observed wavelength dependencies, and more.

**Response:** We are very grateful to the reviewers for their scientific advice, which has been fully considered in the revised version. Following your comments, we have added the following description to clarify the methods and assumptions used for the inversion of the dust extinction profiles by the AD-Net lidar observations.

Lines 278-286: *"In this study, the DEC profile at 532 nm provided by an AD-Net site named "Zamynuud" were used to explore the impacts of long-range dust transport on downstream areas. Each lidar from AD-Net takes a 5-minute measurement every 15 minutes, generating an observation data file with the vertical resolution of 30m. For AD-Net, the Klett's inversion method was employed to derived the extinction coefficient, after applying a geometrical-form-factor correction (Sugimoto et al., 2003). With the assumption of external mixing of dust and spherical aerosols, the ratio (R) of contribution of dust in the extinction coefficient is calculated as follows:*

$$R = \frac{(\delta_a - \delta_2)(1 + \delta_1)}{(1 + \delta_a)(\delta_1 - \delta_2)} \qquad (7)$$

*where $\delta_a$ is the observed aerosol depolarization ratio (ADR). $\delta_1$ and $\delta_2$ are ADRs of dust and air-pollution aerosols. The values of $\delta_1$ and $\delta_2$ were determined empirically and are 0.35 and 0.05, respectively (Sugimoto et al., 2003; Shimizu et al., 2004)."*

[Figure]

*Figure 7: Time–height evolution of dust extinction coefficient (km$^{-1}$) at 532 nm retrieved by ground-based Lidar (location of the site shown in Fig. 1) during the 3.15 and 3.27 events.*

Although similar studies have been performed for the specific SDSs and the study domain, improvements of the present work may lead to very interesting contribution in the literature, due to the very special and diverse conditions encountered over the Eastern part of Asia. At its present state, and taking the above comments under consideration, I suggest to the

journal to reject the paper. However, I would suggest the authors to go through the entire manuscript once more and build on top of the published studies and on this work, follow the suggested improvements/recommendations, and maybe focus on more novel ideas such the effects of the studied SDSs in terms of atmospheric chemistry/physics (i.e. RT, IN/CCN, dust deposition rate), and then I would encourage them to resubmit it.

**Response:** Once again, we sincerely thank the reviewers for these structural suggestions, which have led to a qualitative improvement of our study. Finally, we thank the reviewers for providing us with some potentially novel ideas that will be taken into account in our future studies.

---

## Author Response (AR2)

Thanks very much for the time and efforts that you have put into reviewing the previous version of the manuscript. We really appreciate all your comments and suggestions that have enabled us to improve the manuscript. The following is a point-to-point response to the reviewer's comments. We have studied comments carefully and have made correction which we hope meet with approval. Revised portion are marked in red in the revised paper.

The authors present a variety of information to characterize two major dust storms in the North China region during March 2021, which were exceptional events. They describe the spatial, temporal and vertical evolution of the dust events using satellite dust optical depths and lidar dust extinction profiles, visibility measurements and RGB Himawari imagery. They analyze the optical and radiative properties of the dust using AERONET data from a site in the Beijing region. They use MERRA-2 reanalysis data to identify dust source locations, and analyse the dynamical driving meteorology predominantly using 700hPa reanalysis data. Finally they put the scale of the month's dust events into context by analysing the historical record of the dust loading from satellite data and environmental reanalysis data. The study is of importance since it characterizes these unusual dust events in detail throughout their lifetime from emission, driving meteorology, characteristics in the atmosphere, transport patterns, optical properties, and radiative effects, providing a host of useful information to the aerosol and climate community. The work on classifying the dust events in their historical context is novel and is very important in understanding these properties. Overall the paper is very well written and presented.

The first round of reviewers pointed out several areas of overlap with two previously published papers (Filonchyk (2022) and Liang et al. (2022)). Liang et al. (2022) is a short, bulletin type of publication of 4 pages. Though it touches on some of the areas covered by this publication, the short length of the publication prohibits exploration of the science and data in any detail, as is done in this article. Filonchyk (2022) focuses on ground-based particulate matter concentrations during the 3.15 event, showing some satellite data (mostly different to that used here) to illustrate the links between the ground-based measurements and confirm dust presence. Some analysis of the synoptic situation is presented in Filonchyk (2022), though scope remains for much more detailed analysis and exploration, as aimed for in this article. Thus I think there is ample scope for an article such as this one, fitting for ACP, where the dust properties, evolution and effects, from several data sources, are explored in much detail.

**Response:** We would like to thank the reviewer for the positive comments on our work. These comments have given us more confidence to further improve the quality of this work.**

The authors give a very strong justification of why their article is different to Filonchyk (2022) and Liang et al. (2022) in the response to reviewers. In my view these are strongly justified, and there is plenty of scope for exploring the event(s) in much more detail, and with a stronger link between the satellite data, dust transport and emission patterns, and the meteorology. However, given this strong justification I was surprised that this did not come through more strongly in the article – both in terms of setting the scene in the introduction, and in terms of the analysis of certain data elements. The authors certainly need to expand the introduction to include a more detailed summary of the published work on this event,

setting out how their aims and approaches expand on what has already been done. They have done this in the response to reviewers, but it feels like it has not really made it through to the article itself. Secondly more could have been made of certain avenues of data analysis which are different to those already published – some even pointed out by the authors themselves. I describe these below, but it is up to the editor whether these is necessary to include. They would enhance the article's novelty, rather than be critical to its publication. **Response:** We thank the reviewer for these constructive suggestions. We apologize for not providing a detailed and comprehensive summary of the existing studies in the introduction of the first revision, and for not highlighting new data and methods that differ from those in previous studies, although these were presented in the response to reviewers. In this revision, we have completely addressed these concerns of yours. For detailed revisions, please refer to the following sections.

**General comments:**

1. RGB Himawari imagery - I was surprised to see limited analysis of RGB Himawari imagery, given the response to reviewers. One immense advantage of the RGB imagery is that it's from geostationary satellites, therefore permitting analysis of dust events at higher temporal resolution than possible with polar-orbiters, providing data once a day. Analysis of hourly RGB data can be highly insightful to dust emission and transport mechanisms, and their interactions and influences by weather systems as a function of time. This simply cannot be determined from single-day overpass satellite data. In my opinion further expansion of the imagery along these lines would be highly beneficial.

**Response:** We feel great thanks for your professional suggests. In the revised manuscript, the following three figures and their descriptions have been included to enhance the understanding of the mechanisms of dust emission and transport, and their interactions with weather systems.

"Moreover, the dynamics of dust emissions are mainly regulated by the synoptic systems. The 3-h RGB dust imageries on March 14 clearly show the interaction between the synoptic systems and dust emissions, which is not always evident in the still images (Fig. S7). The dust plume formed at 12:00 CST on March 14 is triggered by the strong wind associated with the movement of the convective system."

"Although the 3-h RGB dust imagery has identified dust activity in south-central Mongolia associated with the incipient Mongolian cyclone at 18:00 CST on March 26 (Fig. S8), the enhancement of dust emissions is mainly controlled by the development and movement of the cyclone on March 27 (Fig. S9)."

Figure S7: The 3-h evolution of dust plumes (magenta) as revealed by Himawari-8 dust RGB composite images on March 14, 2021. Overlaid on the RGB imagery is the 3-h ERA5 wind vectors at 10m.